# The RAS-related GTPase RHOB confers resistance to EGFR-tyrosine kinase inhibitors in non-small-cell lung cancer via an AKT-dependent mechanism

Olivier Calvayrac[1,2,†], Julien Mazières[1,2,3,*,†] ⓘD, Sarah Figarol[1], Claire Marty-Detraves[1], Isabelle Raymond-Letron[4], Emilie Bousquet[3], Magali Farella[1,5], Estelle Clermont-Taranchon[6], Julie Milia[2,3], Isabelle Rouquette[6], Nicolas Guibert[1,3], Amélie Lusque[7], Jacques Cadranel[8], Nathalie Mathiot[8], Ariel Savina[9], Anne Pradines[1,5] & Gilles Favre[1,2,5,**] ⓘD

## Abstract

Although lung cancer patients harboring EGFR mutations benefit from treatment with EGFR-tyrosine kinase inhibitors (EGFR-TKI), most of them rapidly relapse. RHOB GTPase is a critical player in both lung carcinogenesis and the EGFR signaling pathway; therefore, we hypothesized that it could play a role in the response to EGFR-TKI. In a series of samples from EGFR-mutated patients, we found that low RHOB expression correlated with a good response to EGFR-TKI treatment while a poor response correlated with high RHOB expression (15.3 versus 5.6 months of progression-free survival). Moreover, a better response to EGFR-TKI was associated with low RHOB levels in a panel of lung tumor cell lines and in a lung-specific tetracycline-inducible EGFR[L858R] transgenic mouse model. High RHOB expression was also found to prevent erlotinib-induced AKT inhibition *in vitro* and *in vivo*. Furthermore, a combination of the new-generation AKT inhibitor G594 with erlotinib induced tumor cell death *in vitro* and tumor regression *in vivo* in RHOB-positive cells. Our results support a role for RHOB/AKT signaling in the resistance to EGFR-TKI and propose RHOB as a potential predictor of patient response to EGFR-TKI treatment.

**Keywords** RhoB; EGFR; TKI; AKT; resistance

**Subject Categories** Cancer; Pharmacology & Drug Discovery; Respiratory System

## Introduction

Epidermal growth factor receptor (EGFR) is the main oncogenic driver in non-small-cell lung cancer (NSCLC), occurring in 10–50% of the patients depending on their origins. EGFR-tyrosine kinase inhibitors (EGFR-TKI) are widely used to treat metastatic NSCLC in patients bearing EGFR-activating mutations, which are most commonly the L858R point mutation and the deletion of exon 19 (Rosell *et al*, 2012). Sixty–seventy percent of the EGFR-mutated patients respond to therapy, 20% have a stable disease, and 15% develop primary resistance. However, despite this high response rate, all patients usually relapse with a median delay of 12 months. Among the few identified mechanisms of primary resistance, low expression levels of the pro-apoptotic protein BIM have been revealed as a good predictor of non-responsiveness to targeted therapy in EGFR-mutated cells and patients (Faber *et al*, 2011), and similar observations have been reported for the RAS GTPase-activating protein NF1 (de Bruin *et al*, 2014). Many mechanisms of acquired resistance have been elucidated so far, such as the T790M gatekeeper mutation (Nguyen *et al*, 2009), amplification of either the MET oncogene (Engelman *et al*, 2007) or HER2 (Takezawa *et al*, 2012), or epithelial-to-mesenchymal transition (EMT) (Thomson *et al*, 2005). These have led to the development of new-generation drugs (Cross *et al*, 2014) that are given sequentially after EGFR-TKI failure. Resistance mechanisms to EGFR-TKI could be mediated by a bypass reactivation of one or several key proliferation and survival signaling pathways downstream from EGFR, mainly PI3K (phosphatidylinositol 3-kinase)/AKT (Engelman *et al*, 2007), MEK

1 Inserm, Centre de Recherche en Cancérologie de Toulouse, CRCT UMR-1037, Toulouse, France
2 Université Paul Sabatier, Toulouse, France
3 CHU Toulouse, IUCT-Rangueil-Larrey, Service de Pneumologie, Toulouse, France
4 Laboratoire d'Histopathologie, UPS-INP-ENVT, UMS006, Université de Toulouse, Toulouse, France
5 Laboratoire de Biologie Médicale Oncologique, Institut Claudius Regaud, IUCT-Oncopole, Toulouse, France
6 Departement d'Anatomo-Cytopathologie, CHU de Toulouse, IUCT-Oncopole, Toulouse, France
7 Institut Claudius Regaud, IUCT-Oncopole, Bureau des Essais Cliniques, Cellule Biostatistiques, Toulouse, France
8 Sorbonne Universités, UPMC Univ. Paris 06, GRC n°04, Theranoscan, Paris, AP-HP, Hôpital Tenon, Service de Pneumologie, Paris, France
9 Institut Roche, Roche SAS, Boulogne-Billancourt, France
*Corresponding author. Tel: +33 5 67 77 18 37; E-mail: mazieres.j@chu-toulouse.fr
**Corresponding author. Tel: +33 5 31 15 52 01; E-mail: favre.gilles@iuct-oncopole.fr
†These authors contributed equally to this work

(mitogen-activated protein kinase kinase)/ERK (extracellular signal-regulated kinase) (Ercan *et al*, 2012), or STAT (signal transducer and activator of transcription) pathways (Lee *et al*, 2014), among others. To date, there is no approved predictive biomarker able to predict tumor sensitivity to EGFR-TKI since most of the resistance mechanisms are acquired during tumor treatment.

RHOB is a RAS-related monomeric GTPase that displays tumor suppressor activity in NSCLC. RHOB expression is downregulated in aggressive tumors, and we have shown that the loss of RHOB is associated with decreased overall survival in two independent series of patients (Calvayrac *et al*, 2014). More recently, we showed that RHOB levels are not only a strong prognostic factor for NSCLC but that its downregulation is also critical for the acquisition of an aggressive phenotype of adenocarcinoma in an EGFR$^{L858R}$-induced tumor model in mice (Calvayrac *et al*, 2014). The underlying mechanism of RHOB-mediated tumor suppression is only partially understood; however, we have shown that RHOB controls cell survival and invasion through PP2A-mediated AKT dephosphorylation (Bousquet *et al*, 2009). RHOB is an early response gene induced by several cell stresses such as genotoxic agents (Fritz *et al*, 1995; Canguilhem *et al*, 2005; Mamouni *et al*, 2014), hypoxia (Skuli *et al*, 2006), and growth factors such as EGF (de Cremoux *et al*, 1994). RHOB has also been shown to regulate the intracellular trafficking of several signaling proteins such as EGFR, PDGFR, SRC, or AKT (Ellis & Mellor, 2000; Adini *et al*, 2003; Sandilands *et al*, 2004). These effects are specific to RHOB since RHOA and RHOC, the most closely related RHO proteins, do not control these responses. In fact, we and others have previously demonstrated that RHOB prevents the trafficking of EGFR between endocytic compartments (Gampel *et al*, 1999), causing the phosphorylated form of EGFR to persist at the plasma membrane and sustain EGFR-dependent AKT signaling (Canguilhem *et al*, 2005; Lajoie-Mazenc *et al*, 2008). This suggests that RHOB expression levels determine the efficacy of EGFR signaling and led us to hypothesize that RHOB levels could account for the initial sensitivity of tumor cells to EGFR-TKI through a mechanism that may involve EGFR-dependent AKT signaling.

We first investigated this hypothesis in patients carrying mutated EGFR who had been treated with EGFR-TKI and demonstrated that RHOB tumor tissue levels predicted patient response rate to EGFR-TKI therapy. We then analyzed the consequences of modulating RHOB levels on EGFR signaling using dedicated cell lines and a mouse model of inducible lung-specific EGFR$^{L858R}$-driven tumors (Politi *et al*, 2006). The results presented here demonstrate that RHOB expression is predictive of EGFR-TKI response and suggest that an EGFR-TKI–AKT inhibitor combination may provide a clinical advantage to prevent resistance to EGFR-TKI in RHOB-positive tumor patients.

## Results

### RHOB expression predicts the response to EGFR-TKI in patients harboring EGFR-activating mutations

We first determined whether RHOB expression in primary lung tumors is predictive of the response to EGFR-TKI in EGFR-mutated patients. We performed RHOB immunohistochemistry analysis on 96 lung tumor biopsies collected before any treatment from a series of EGFR-mutated lung adenocarcinoma. Patients received EGFR-TKI

treatment (erlotinib, $n = 43$; gefitinib, $n = 51$; afatinib, $n = 2$) as first-line ($n = 63$), second-line ($n = 28$), third-line ($n = 3$), or fourth-line ($n = 2$) therapy. According to the intensity of the staining, we defined four levels: null: 0; weak: +; moderate: ++; and high: +++ (Fig 1A). Samples with null or weak staining were considered as low-RHOB patients and samples with moderate or high staining as high-RHOB patients (Fig 1A). Tumor tissues and completed follow-up files were available for all these patients (Fig 1B and Appendix Table S1). Median progression-free survival (PFS) was 12.06 months (95% CI [8.11; 13.99]) for the whole population (Fig EV1A). We observed an impressive clinical response to EGFR-TKI in patients with low RHOB expression, suggesting that RHOB could predict EGFR-TKI sensitivity, as exemplified in Fig 1C. PFS was not statistically different between the RHOB (0) and RHOB (+) groups and between the RHOB (++) and RHOB (+++) groups (Fig EV1B), allowing us to group RHOB (0) with (+) and RHOB (++) with (+++). This suggests that there is a RHOB threshold that defines the EGFR-TKI response, with a clear cutoff between weak and moderate RHOB expression. Indeed, median PFS was 15.3 months (95% CI [13.1; 18.2]) for patients with low RHOB expression (0 and +) and 5.6 months (95% CI [3.6; 6.4]) for patients with high RHOB expression (++ and +++) ($P < 0.0001$; Fig 1D). As an alternative method, we determined RHOB mRNA expression in a subset of tumors, and we observed a significant correlation between RHOB mRNA expression (determined by RT–qPCR) and RHOB protein staining (determined by IHC) (Appendix Fig S1A–C). PFS analysis showed that both methods of detection gave similar results on the predictive role of RHOB in response to EGFR-TKI (9.0 months [1.6; 13.7] versus 14.0 months [5.7; 27.1], $P = 0.0643$, for mRNA; and 9.0 months [5.8; 13.7] versus 20.6 months [12.3; 27.2], $P = 0.0059$, for IHC; Appendix Fig S1D and E). Univariate analysis showed that TKI type was significantly associated with PFS but not age, sex, number of previous lines of chemotherapy, or type of EGFR mutation (Appendix Table S1). We realized a multivariate analysis to demonstrate that RHOB is an independent prognostic factor of PFS ($P < 0.001$; Appendix Table S2). We then analyzed RHOB expression on tumor biopsies of patients receiving EGFR-TKI as a first-line (Fig 1E) or as second- to fourth-line (Fig 1F) therapy. Importantly, high RHOB expression was strongly associated with a shorter PFS to EGFR-TKI, regardless of the line of EGFR-TKI therapy, indicating that the predictive value of RHOB was not affected by previous chemotherapeutics treatments. Moreover, we tested RHOB expression on tumor biopsies of 11 patients at the moment of diagnostic and after relapse. As shown in Fig 1G, RHOB expression increased at the moment of relapse in eight patients, one patient had no change, and two patients showed RHOB decrease. These results are consistent with the role of RHOB in EGFR-TKI resistance.

Altogether, these results demonstrate that high RHOB might be driving EGFR-TKI resistance in EGFR-mutated patients and that the determination of RHOB expression in the biopsy at the diagnostic could be a robust predictive biomarker of responsiveness to anti EGFR therapy.

### EGFR$^{L858R}$ *Rhob*-positive mouse tumors are resistant to erlotinib, whereas EGFR$^{L858R}$ *Rhob*-deficient tumors are sensitive

We then investigated whether RHOB expression influences the response to EGFR-TKI in a *Rhob*-deficient mouse model. We used a

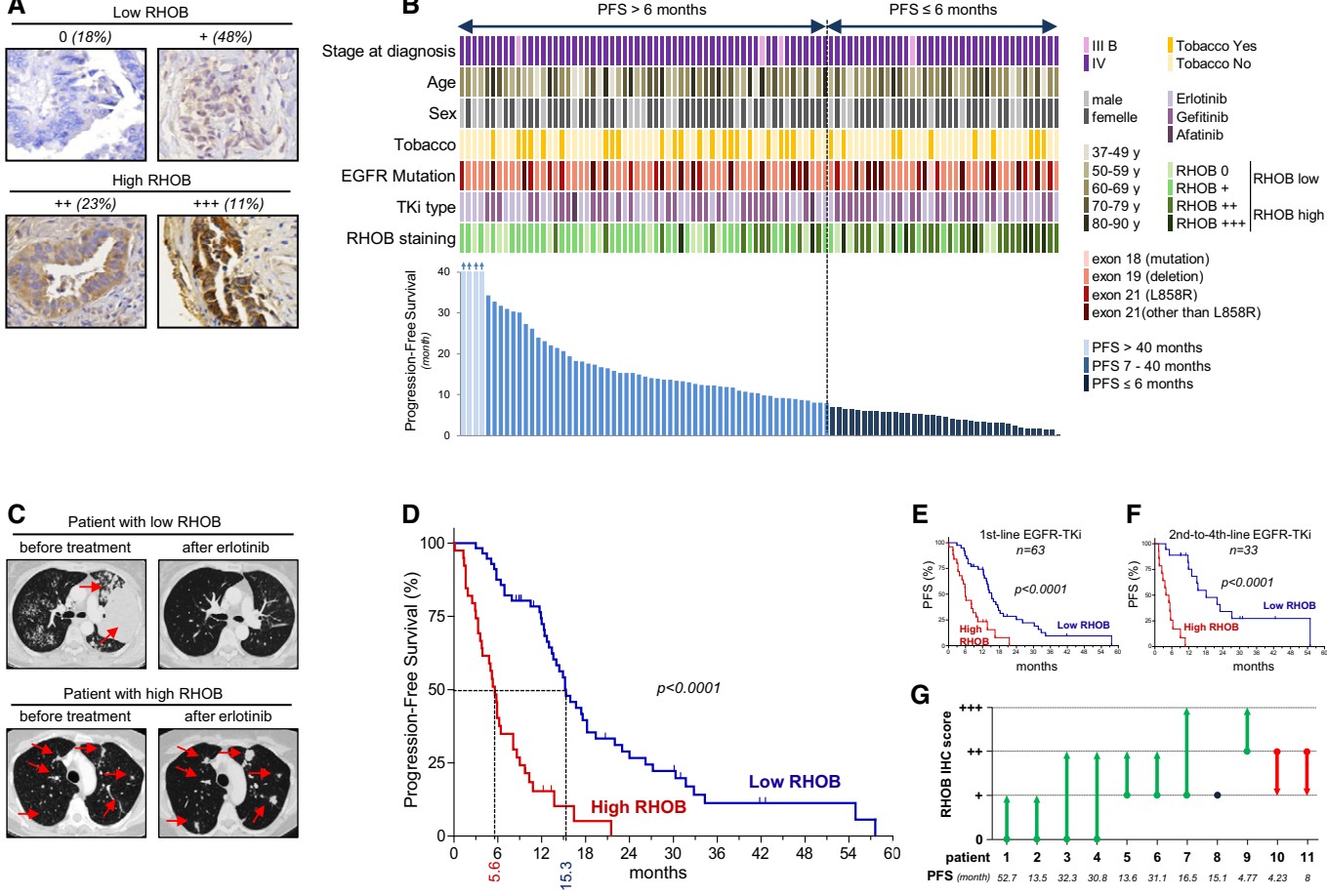

**Figure 1. RHOB expression predicts the response to EGFR-TKI in patients harboring EGFR-activating mutations.**

A   Four representative images of RHOB immunostaining in human non-small-cell lung cancers. Figures correspond to the percentage of tumors analyzed.

B   Characteristics of 96 patients with an EGFR-activating mutation who were treated with EGFR-TKI. Each column corresponds to one patient. Data include general characteristics (stage at diagnosis, age, sex, tobacco usage, position of EGFR mutation, TKI type), progression-free survival (PFS; lower graph), and intensity of RHOB staining as determined by immunohistochemistry (null: 0; weak: +; moderate: ++; and high: +++).

C   Representative scans of patients with low- or high-RHOB-expressing lung tumors, before and after erlotinib treatment. Red arrows indicate lung tumors.

D   Progression-free survival of erlotinib-treated patients with EGFR-mutated lung tumors, according to RHOB expression, assessed by immunohistochemistry (low-RHOB group = negative + weak staining; high-RHOB group = moderate + high staining). *P*-values were determined by the Kaplan–Meier method.

E   Progression-free survival of patients who received EGFR-TKI as first-line therapy (*n* = 63). *P*-values were determined by the Kaplan–Meier method.

F   Progression-free survival of patients who received EGFR-TKI as second-line (*n* = 28), third-line (*n* = 3), or fourth-line (*n* = 2) therapy. *P*-values were determined by the Kaplan–Meier method.

G   RHOB immunostaining score evolution in EGFR-mutated lung tumors before treatment and after EGFR-TKI relapse.

previously described mouse model of inducible lung-specific EGFR$^{L858R}$-driven tumors crossed into a *Rhob* wild-type, heterozygous, or null genetic background (Calvayrac *et al*, 2014). To evaluate the effect of RHOB loss on erlotinib sensitivity, we performed a 4-day treatment with erlotinib at 12.5 mg/kg/day that did not show objective response in the initially described mouse model (Politi *et al*, 2006). In these conditions, erlotinib treatment induced a strong anti-tumoral response in EGFR$^{L858R}$/*Rhob*$^{-/-}$ and EGFR$^{L858R}$/*Rhob*$^{+/-}$ mice, whereas no significant tumor shrinkage was observed in EGFR$^{L858R}$/*Rhob*$^{+/+}$ mice (Fig 2A). Erlotinib induced a significant decrease in tumor/total lung surface ratios in *Rhob*-deficient and heterozygous mice (Fig 2B), and lungs of these mice were almost clear of tumor cells after treatment. In addition, cell proliferation, measured by the Ki67-positive cell ratio, decreased in

*Rhob*$^{+/-}$ and *Rhob*$^{-/-}$ mice but not in *Rhob*$^{+/+}$ mice (Fig 2C and D). Moreover, caspase-3 cleavage was detected after 24 h of treatment in EGFR$^{L858R}$/*Rhob*$^{-/-}$ but not in EGFR$^{L858R}$/*Rhob*$^{+/+}$ mice, suggesting a strong and rapid apoptotic response to erlotinib in *Rhob*-deficient mice (Fig 2E). These observations with EGFR-mutated driven lung tumors in mice confirm our data obtained with patients, showing that low RHOB expression is associated with a good response to EGFR-TKI.

**The modulation of RHOB expression determines the level of resistance to erlotinib in EGFR-mutated lung cancer cell lines**

To further confirm that RHOB expression impacts on the response to EGFR-TKI treatment, we downregulated or

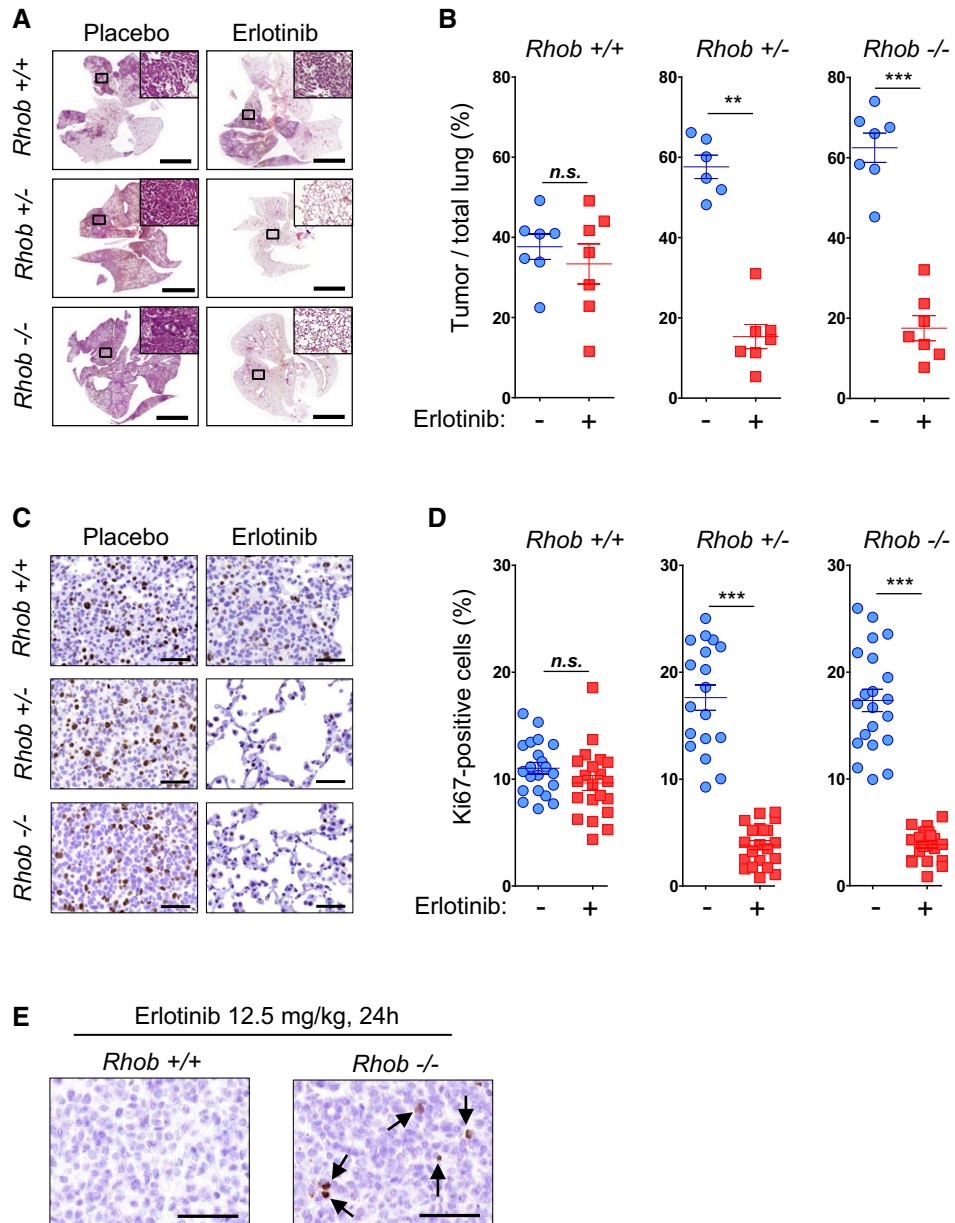

**Figure 2. RHOB loss of expression increases sensitivity to erlotinib in mice with EGFR^L858R-driven lung tumors.**

A    Representative H&E staining of whole lungs from *EGFR^L858R/Rhob^−/−*, *EGFR^L858R/Rhob^+/−*, and *EGFR^L858R/Rhob^+/+* mice treated or not with erlotinib (12.5 mg/kg/day) for 4 days. Scale bars: 5 mm.

B    Quantification of the tumor/lung ratio. *n* = 7 for each group except for *EGFR^L858R/Rhob^+/−* placebo (*n* = 6).

C, D    Representative Ki67 immunostaining of *EGFR^L858R/Rhob^−/−*, *EGFR^L858R/Rhob^+/−*, and *EGFR^L858R/Rhob^+/+* mice treated or not with erlotinib (12.5 mg/kg/day) for 4 days (scale bars: 50 μm), and the corresponding quantification (D). Three independent zones per mouse lung were used for quantification. *n* = 21 (seven mice) for each group except for *EGFR^L858R/Rhob^+/−* placebo (*n* = 18; six mice).

E    Immunostaining of cleaved caspase-3 in lung tumors from *EGFR^L858R/Rhob^+/+* or *EGFR^L858R/Rhob^−/−* mice treated for 24 h with erlotinib at 12.5 mg/kg. Black arrows point apoptotic cells. Scale bars: 50 μm.

Data information: **$P < 0.001$ versus placebo; ***$P < 0.0001$ versus placebo. Data are expressed as mean ± SEM, *P*-values were determined by Mann–Whitney two-tailed *t*-test.

overexpressed RHOB by RNA interference and adenoviral transduction, respectively, in several EGFR-mutated human lung cancer cell lines. Interestingly, and in accordance with our previous results, RHOB downregulation hypersensitized HCC4006 cells

to erlotinib (Fig 3A) while its overexpression protected them (Fig 3B). To avoid any possible siRNA off-target effects, siRNA-transfected cells were transduced with a RHOB-expressing adenovirus. Figure 3C shows that RHOB overexpression completely

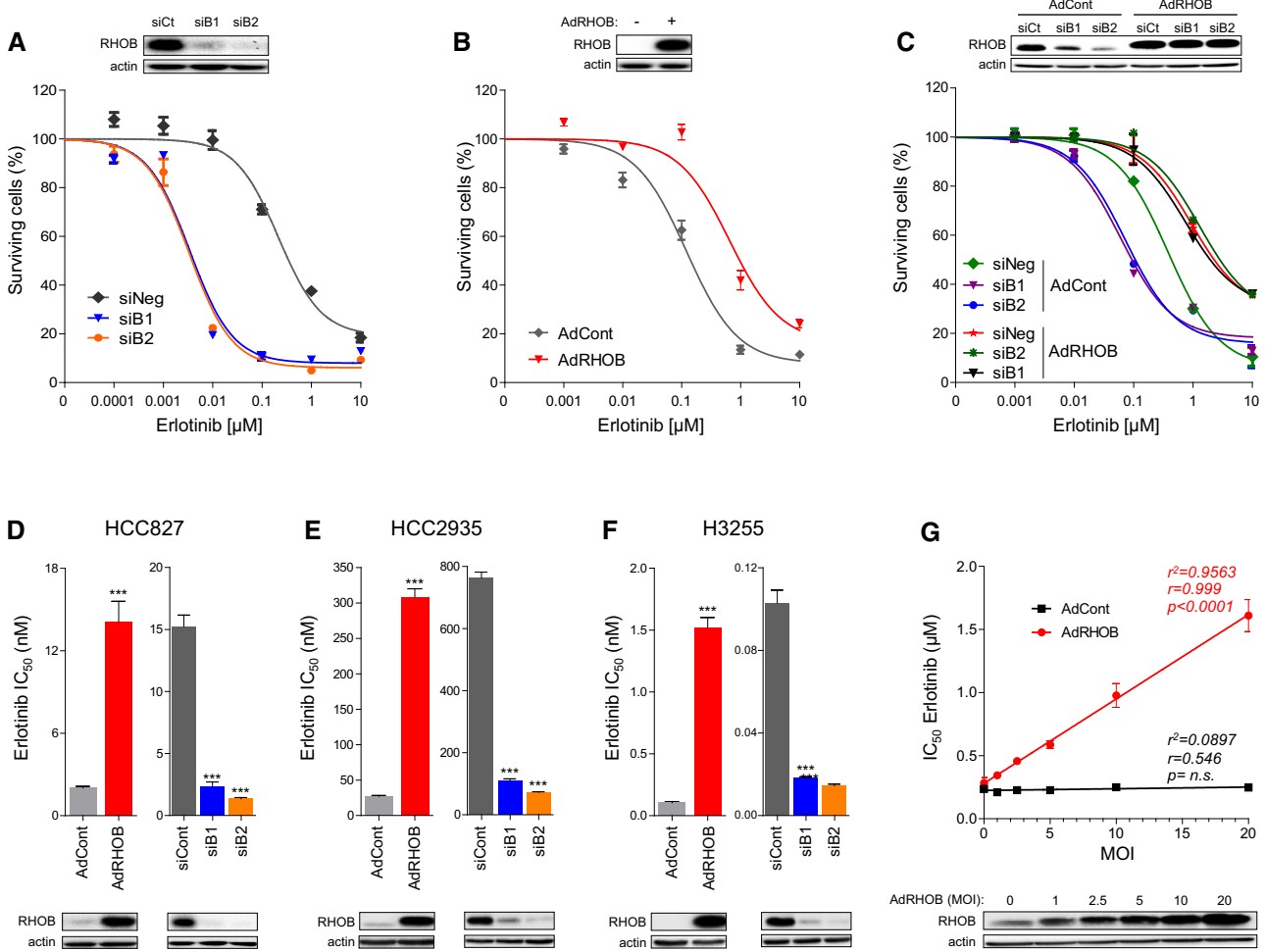

**Figure 3. The modulation of RHOB expression determines the level of resistance to erlotinib in EGFR-mutated lung cancer cell lines.**

A–C  HCC4006 cells were either (A) transfected with two siRNA against RHOB (siB1, siB2), (B) transduced with control (AdCont) or RHOB-overexpressing adenoviruses (AdRHOB), or (C) both transfected with siRNA and transduced by adenoviruses, and then treated with increasing doses of erlotinib. The surviving cell fraction was determined by an MTS assay after 72 h and compared to untreated cells.

D–F  Erlotinib IC$_{50}$ values were quantified in RHOB-overexpressing or RHOB-depleted (D) HCC827 cells, (E) HCC2935 cells, and (F) H3255 cells, as determined by an MTS assay after 72-h treatment. RHOB overexpression or inhibition was monitored by Western blotting for each condition.

G  HCC4006 cells were transduced with control (AdCont) or RHOB-overexpressing adenoviruses (AdRHOB) at an increasing multiplicity of infection (MOI); then, erlotinib IC$_{50}$ values were determined after 72 h by an MTS assay, and a correlation analysis was performed. RHOB overexpression was monitored by Western blotting.

Data information: ***$P < 0.0001$ versus control cells. Data are representative of at least three independent experiments. Data are expressed as mean ± SEM, *P*-values were determined by unpaired two-tailed Student's *t*-test.

Source data are available online for this figure.

reversed the effect of RHOB downregulation on erlotinib sensitivity. Similar results were obtained with several cell lines harboring either an exon 19 deletion (HCC827 and HCC2935) (Figs 3D and E, and EV2A–C) or an exon 21 L858R point mutation (H3255) (Figs 3F and EV2D), which together account for more than 85% of all activated EGFR mutations found in human lung cancers. Moreover, a gradual increase in RHOB expression, obtained through RHOB recombinant adenoviral transduction, gradually increased the erlotinib IC$_{50}$ values in HCC4006 (Fig 3G) and HCC827 cells (Appendix Fig S2). These findings clearly demonstrate that RHOB expression levels determine the resistance of EGFR-mutated cells to erlotinib.

## RHOB-induced resistance to erlotinib involves the AKT pathway

To investigate the mechanisms underlying the involvement of RHOB in the resistance to erlotinib treatment, we analyzed the potential role of RHOB in cell survival. We first tested a panel of EGFR-mutated cell lines for their ERK and AKT phosphorylation status, two key actors of the EGFR signaling pathway involved in the sensitivity of tumor cells to EGFR-TKI (Niederst & Engelman, 2013). The cell lines were transduced with an adenovirus control or with a RHOB recombinant adenovirus. As expected, in EGFR-mutated, but not in EGFR WT, cell lines, erlotinib inhibited EGFR, ERK, and AKT phosphorylation in each of the control cells (Figs 4A and EV3). In

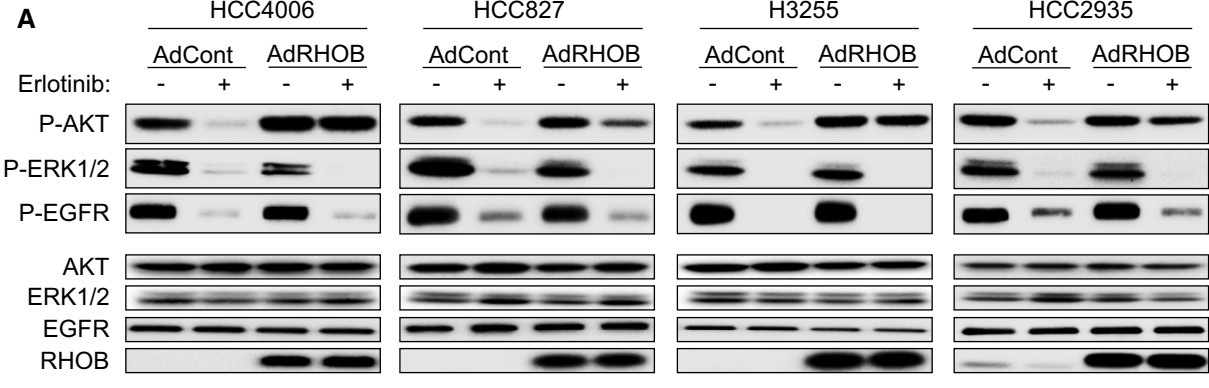

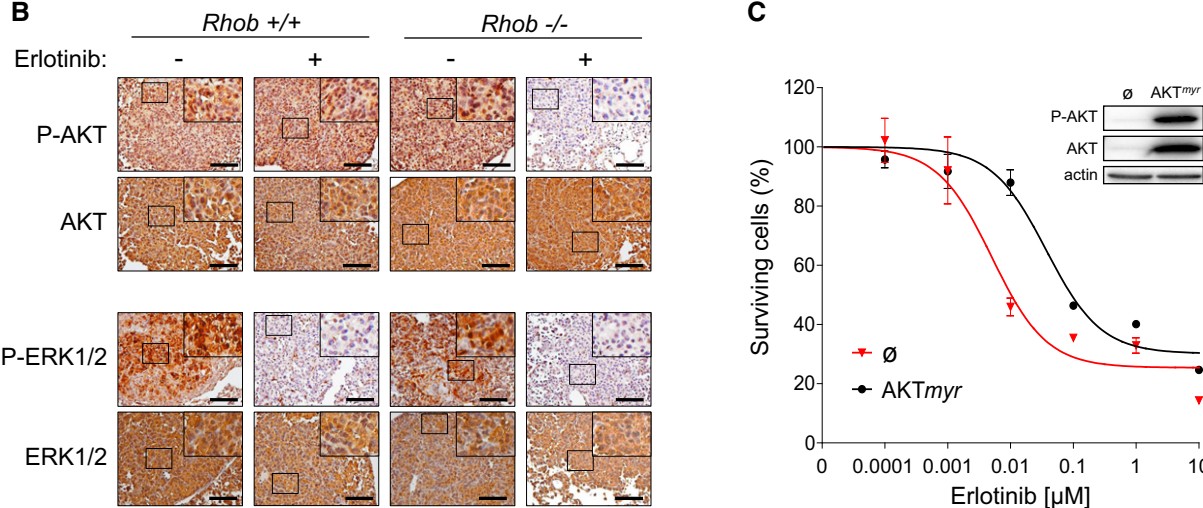

**Figure 4. RHOB induces resistance to erlotinib through the AKT pathway.**

A  HCC4006, HCC827, HCC2935, and H3255 cells were transduced with control (AdCont) or RHOB-overexpressing (AdRHOB) adenoviruses and treated for 4 h with erlotinib at concentrations corresponding to the respective $IC_{50}$ values determined for each control cell line. The phosphorylation status of AKT, ERK1/2, and EGFR was assessed by Western blotting and normalized according to total protein levels. RHOB overexpression was also monitored by Western blotting.

B  Representative immunostaining of phospho-AKT (Ser473) and phospho-ERK1/2 and their total protein amounts in lung tumors from $EGFR^{L858R}/Rhob^{-/-}$ or $EGFR^{L858R}/Rhob^{+/+}$ mice treated or not with erlotinib (12.5 mg/kg/day) for 4 days. The remaining hyperplastic areas were selected in erlotinib-treated mice to efficiently characterize the effect of erlotinib on ERK and AKT pathways in both *Rhob* genotypes. Scale bars: 100 μm.

C  HCC4006 cells were transfected with a plasmid coding for a constitutively active AKT mutant (AKT*myr*, myristoylated) or an empty vector (ø) and treated for 72 h with increasing concentrations of erlotinib. The surviving cell fraction was determined by an MTS assay, and AKT overexpression and phosphorylation at Ser473 were assessed by Western blotting. Data are representative of at least three independent experiments. Data are expressed as mean ± SEM from three independent experiments.

Source data are available online for this figure.

contrast, when RHOB was overexpressed in the four EGFR-mutated cell lines, EGFR and ERK phosphorylation was still downregulated by erlotinib whereas AKT remained fully phosphorylated (Fig 4A), and no effect was observed on EGFR WT cells (Fig EV3). Consistent with this, erlotinib more potently inhibited AKT phosphorylation in $Rhob^{-/-}$ lung tumor mice compared to $Rhob^{+/+}$ mice, while erlotinib inhibited ERK phosphorylation in both mouse models (Fig 4B). These results demonstrate that RHOB prevents erlotinib-induced AKT dephosphorylation, suggesting that the RHOB/AKT axis might account for the sensitivity of EGFR-mutated tumor cells to EGFR-TKI treatment. In agreement with this hypothesis, we found that the

constitutive expression of AKT*myr,* an active AKT mutant, phenocopied the effect of RHOB overexpression onto HCC4006 (Fig 4C) and HCC827 cells (Appendix Fig S3) in terms of increasing their erlotinib $IC_{50}$ values.

### AKT inhibition reverses RHOB-induced resistance to erlotinib-treated cell lines

We next investigated whether AKT inhibition could reverse RHOB-induced resistance to erlotinib, using the novel selective AKT inhibitor G594 (a primary compound of GDC-0068, ipatasertib) (Lin

et al, 2013). Firstly, we confirmed that G594 induced AKT inhibition by monitoring the phosphorylation of its substrate GSK3β. Indeed, G594 induced a previously described AKT phosphorylation at both Thr[308] and Ser[473] residues (Blake et al, 2012; Lin et al, 2013) while

inhibiting its activity. As expected, we evidenced that G594 prevented GSK3β phosphorylation in RHOB-overexpressing HCC4006 cells treated with erlotinib (Fig 5A). We then tested the effects of G594 on cell survival in control or RHOB-overexpressing

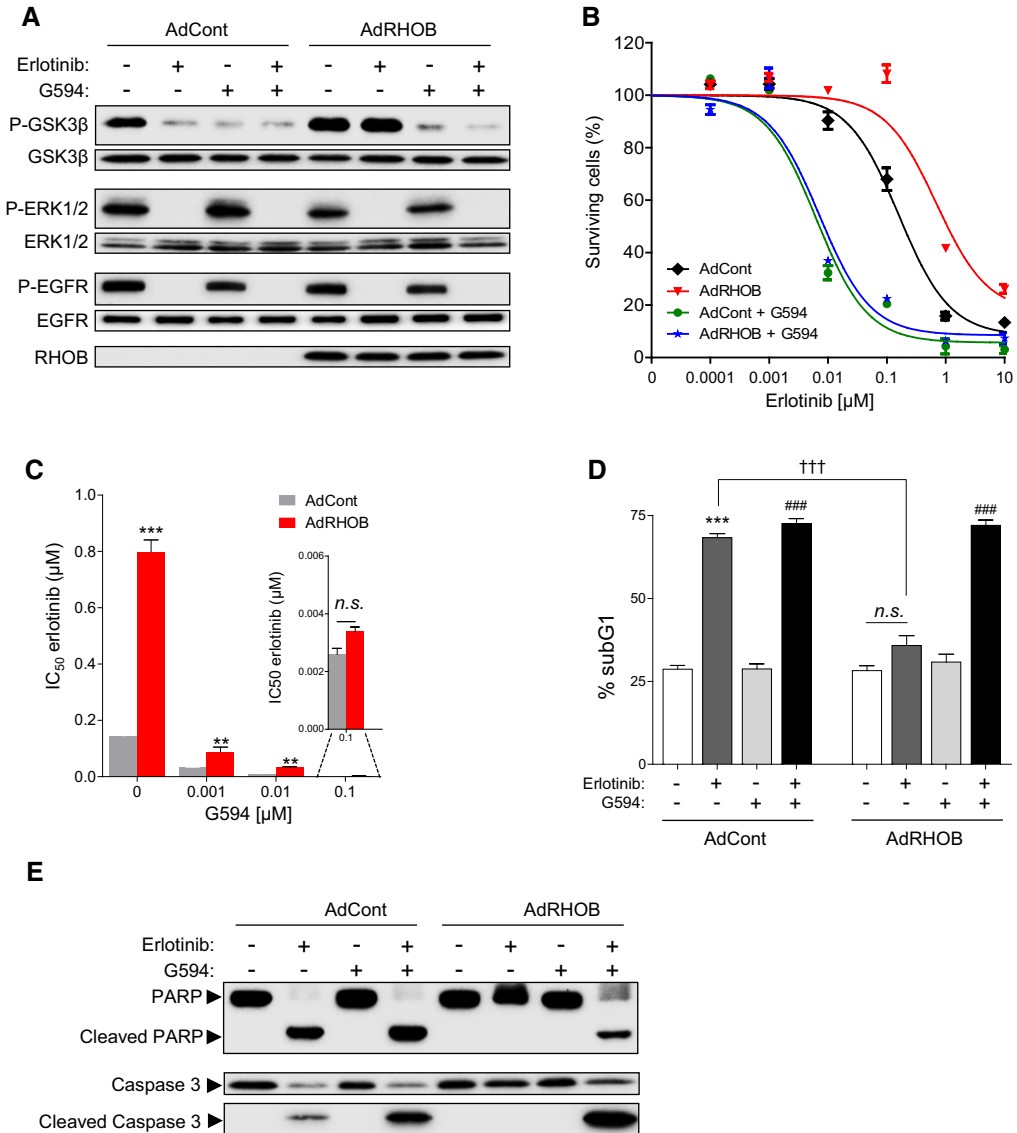

**Figure 5. AKT inhibition sensitizes RHOB-expressing cells to erlotinib.**

A    HCC4006 cells were transduced with control (AdCont) or RHOB-overexpressing (AdRHOB) adenoviruses and treated for 4 h with erlotinib (100 nM), G594 (100 nM), or a combination of both drugs. The phosphorylation status of GSK3β (Ser9), ERK1/2, and EGFR (Tyr1173) was assessed by Western blotting and normalized according to the total protein levels. RHOB overexpression was also monitored by Western blotting.

B    HCC4006 cells were transduced with control (AdCont) or RHOB-overexpressing (AdRHOB) adenoviruses and treated for 72 h with erlotinib alone (black and red curves) or in combination with the AKT inhibitor G594 at 100 nM (green and blue curves). The surviving cell fraction was determined by an MTS assay. Data are expressed as mean ± SEM from three independent experiments.

C–E    HCC4006 cells were transduced with control (AdCont) or RHOB-overexpressing adenoviruses (AdRHOB) and treated for 72 h with increasing concentrations of erlotinib in the absence or presence of increasing doses of G594. The surviving cell fraction was determined by an MTS assay, and erlotinib IC50 values were determined for each condition (**P < 0.001 versus AdCont cells; ***P < 0.0001 versus AdCont cells). HCC4006 cells were transduced with control (AdCont) or RHOB-overexpressing (AdRHOB) adenoviruses and treated for 48 h with erlotinib (100 nM), G594 (100 nM), or a combination of both drugs. Apoptosis was then determined by either quantification of the subG1 cell population (D) or detection of cleaved PARP and caspase-3 (E) (***P < 0.0001 versus untreated cells; ###P < 0.0001 versus G594 treated cells; †††P < 0.0001 versus erlotinib-treated AdCont cells). In vitro data are representative of at least three independent experiments. Data are expressed as mean ± SEM from three independent experiments, P-values were determined by unpaired two-tailed Student's t-test.

Source data are available online for this figure.

HCC4006 cells treated with erlotinib. G594 significantly decreased erlotinib $IC_{50}$ values in RHOB-overexpressing cells, suggesting that AKT inhibition reversed RHOB-induced resistance to erlotinib (Fig 5B). Interestingly, this effect was observed at concentrations of G594 as low as 1 nM in HCC4006 (Fig 5C) and 10 nM in HCC827 cells (Appendix Fig S4), concentrations at which cell survival was not affected (Appendix Fig S5). We tested a panel of three other EGFR-mutated cell lines and demonstrated that G594 had similar effects on both AKT inhibition (Fig EV4A–C) and the reversal of RHOB-induced erlotinib resistance (Fig EV4D–F).

Lastly, we investigated the mechanism by which the combination of erlotinib and G594 inhibited cell survival. We tested a panel of EGFR-mutated cell lines for the induction of apoptosis, characterized by the percentage of subG1 cells, as well as cleavage of PARP and caspase-3. Erlotinib induced a significant increase in proportion of cells in subG1 phase and PARP and caspase-3 cleavage in control sensitive cells, but not in RHOB-overexpressing cells, suggesting that RHOB-overexpressing cells are resistant to erlotinib-induced apoptosis (Figs 5D and E, and EV5). These findings are consistent with the mouse model results shown in Fig 2E. Altogether, these data indicate that AKT inhibition reverses RHOB-induced resistance to erlotinib and strongly suggest that RHOB triggers resistance to erlotinib through AKT activation.

### *In vivo* AKT inhibition reverses RHOB-induced resistance to erlotinib in EGFR[L858R] mice

To validate the above findings, we investigated whether AKT inhibition would reverse RHOB-induced resistance to erlotinib *in vivo*. We tested the effect of erlotinib in combination with G594 in the mouse models described in Fig 2. We determined the tumor/total lung ratios and the proliferating index, revealed by Ki67 staining. G594 treatment alone had no effect on $Rhob^{+/+}$ mice, but it induced a significant decrease in the tumor/total lung ratio (Fig 6A and B) and in Ki67-positive cells in $Rhob^{-/-}$ (Fig 6C and D). We also observed a significant decrease in the tumor/total lung and the Ki67-positive cell ratios in $Rhob^{+/+}$ mice treated with the combination of erlotinib and G594 compared to the individual treatments (Fig 6). Interestingly, the combination of the two drugs caused the two parameters to reach the same values as the heterozygous or *Rhob*-invalidated mice treated with erlotinib as a single agent. In addition, we observed no difference in the tumor/total lung ratios in $Rhob^{+/-}$ and $Rhob^{-/-}$ mice treated with the combination of drugs. These data demonstrated that G594 is a potent agent that can resensitize EGFR[L858R]/$Rhob^{+/+}$-resistant mice to erlotinib.

## Discussion

Lung cancer patients have benefited from targeted therapy in the last decade, providing new hope in the management of advanced NSCLCs. EGFR-TKI such as erlotinib (Rosell *et al*, 2012), gefitinib (Mok *et al*, 2009), and afatinib (Sequist *et al*, 2013) have shown clinical activity toward NSCLC, leading to their approval for the treatment of metastatic disease. However, although seventy percent of patients that harbor EGFR-mutated lung tumors respond to EGFR-TKI, almost all develop irremediable resistance mechanisms.

The major goals for increasing treatment success rates in these patients are to improve the initial response to EGFR-TKI and to postpone disease recurrence. Here, our findings demonstrate that a high level of RHOB protein expression in the primary tumor impairs the response rate through a mechanism involving AKT. In fact, AKT inhibition reverses EGFR-TKI resistance in cells with high levels of the RHOB protein. These results have led us to propose a combination of EGFR-TKI and AKT inhibitor as treatment to overcome the primary resistance to EGFR-TKI in RHOB-positive patients.

The interaction of AKT with RHOB seems to be dependent on the cellular context. We and others have shown that the loss of RHOB expression is able to activate AKT (Bousquet *et al*, 2009, 2016) but can also sustain AKT activation in endothelial cells after angiogenic switching (Kazerounian *et al*, 2013). In lung cancer cells, we recently demonstrated that RHOB downregulation decreases PP2A activity, limiting AKT dephosphorylation and maintaining a high level of AKT activation. This suggests that AKT inhibition favors antitumor activity in RHOB-deficient cells. In line with this hypothesis, G594 treatment induced tumor regression in RHOB-deficient but not in wild-type mice. Together this suggests that tumor RHOB levels could determine the response to AKT inhibitor therapy when it is administered as a single agent.

Interestingly, our *in vitro* and *in vivo* results strongly suggest that RHOB is critical for both tumor growth and the apoptotic response to erlotinib, by preventing erlotinib-induced AKT dephosphorylation and leading to the maintenance of a high level of active AKT. It has been shown that RHOB can delay the intracellular trafficking of EGFR (Gampel *et al*, 1999) and restrict EGFR cell surface occupancy (Kazerounian *et al*, 2013), thus modifying EGFR-dependent downstream signaling (Canguilhem *et al*, 2005; Lajoie-Mazenc *et al*, 2008). Our results add to this by showing that RHOB can modify AKT but not ERK signaling in response to erlotinib.

The PI3K/AKT pathway is known to control the oncogenic addiction observed in EGFR-mutated lung cancer, and its activation has been shown to be a crucial event in the resistance to targeted therapies (Obenauf *et al*, 2015). RHOB, through its ability to prevent AKT inhibition, appears to be a key player in the failure of targeted therapies. Importantly, our results from the analysis of RHOB expression in a series of samples from EGFR-mutated lung adenocarcinoma patients treated with EGFR-TKI showed that sensitivity to EGFR-TKI treatment is strikingly higher in patients presenting low RHOB levels, while patients with high RHOB levels had a stronger resistance to treatment. These observations clearly suggest that RHOB expression could be used as a marker in the clinic to predict the efficacy of these kinds of therapeutic strategies, which might represent a clear advantage when making decisions on alternative treatment. Moreover, RHOB predictive value is not modified by treatments given between initial assessment and EGFR-TKI treatment. RHOB high expression may thus help clinicians to anticipate frequent and rapid resistance to EGFR-TKI. The question of the variable abundance of RHOB in lung tumors is not yet fully understood. In a TCGA-based analysis using the lung adenocarcinoma database, no RHOB mutation or deletion was found (Appendix Fig S6A), which confirms previous observations in lung cancer (Sato *et al*, 2007) and in our laboratory (unpublished data) or in other cancers such as head and neck carcinoma (Adnane *et al*, 2002) or breast cancer (Fritz *et al*, 2002). RHOB mRNA levels were not significantly altered by copy number variations or major oncogenic mutations (Appendix Fig S6B–D), but

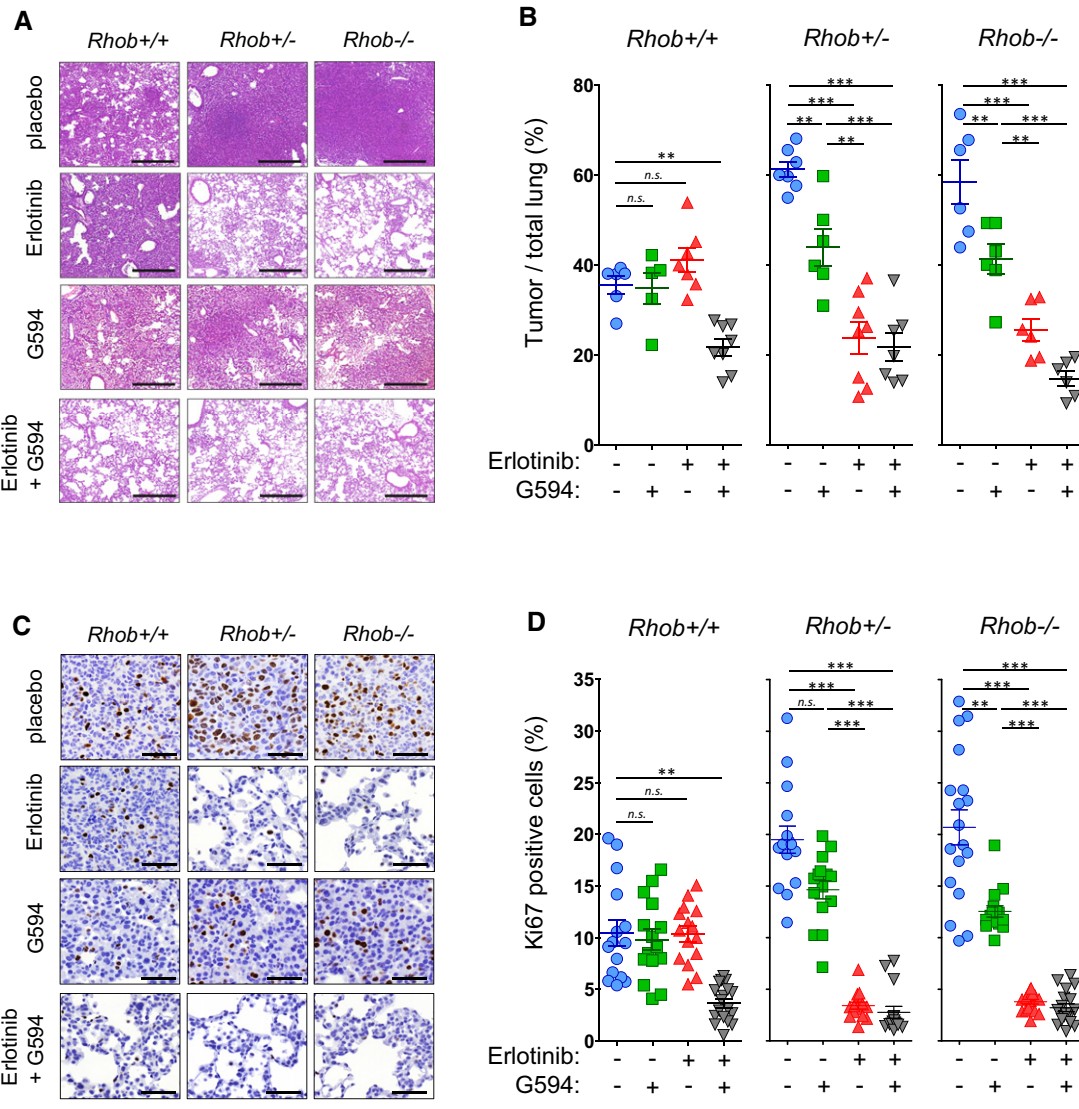

**Figure 6. The AKT inhibitor G594 resensitizes EGFR^L858R mouse tumors to erlotinib.**

A    Representative H&E staining of lung tumors from *EGFR^L858R/Rhob^−/−*, *EGFR^L858R/Rhob^+/−*, and *EGFR^L858R/Rhob^+/+* mice treated or not during 4 days with erlotinib (12.5 mg/kg/day), the AKT inhibitor G594 (25 mg/kg/day), or a combination of both drugs. Scale bars: 500 μm.

B    Quantification of the tumor/lung ratio of mice treated or not with the individual drugs or with a combination of both.

C, D    Representative Ki67 immunostaining of lung tumors from *EGFR^L858R/Rhob^−/−*, *EGFR^L858R/Rhob^+/−*, and *EGFR^L858R/Rhob^+/+* mice treated or not with erlotinib alone (12.5 mg/kg/day) or in combination with the AKT inhibitor G594 (25 mg/kg/day) for 4 days (scale bars: 50 μm), and the corresponding quantification (D).

Data information: $**P < 0.001$; $***P < 0.0001$. *EGFR^L858R/Rhob^+/+* (placebo: $n = 6$; G594: $n = 5$; erlotinib: $n = 7$; G594 + erlotinib: $n = 8$); *EGFR^L858R/Rhob^+/−* (placebo: $n = 7$; G594: $n = 6$; erlotinib: $n = 8$; G594 + erlotinib: $n = 7$); *EGFR^L858R/Rhob^−/−* (placebo: $n = 6$; G594: $n = 6$; erlotinib: $n = 6$; G594 + erlotinib: $n = 6$). Three independent zones per mouse lung were used for Ki67 quantification. Data are expressed as mean ± SEM, *P*-values were determined by Mann–Whitney two-tailed *t*-test.

seemed to be rather associated with epigenetic mechanisms involving particularly miR-21 expression (Appendix Table S3), a well-known oncomir, and chromatin acetylation (Appendix Fig S7), in accordance with previous observations (Wang *et al*, 2003; Mazieres *et al*, 2007; Sato *et al*, 2007; Connolly *et al*, 2010).

Remarkably, by using mouse models we provide evidence that RHOB-induced resistance can be reversed by AKT inhibition. We clearly show that a combination of AKT inhibitors with EGFR-TKI can sustain the sensitivity of RHOB-overexpressing cells to EGFR-TKI. Interestingly, neither AKT inhibitors nor EGFR-TKI induce significant cell death by themselves, but in combination they are

highly potent at inducing apoptosis, suggesting that their combined effects have lethal consequences for tumor cells. AKT inhibitors are currently under clinical development with promising results. Ipatasertib, which corresponds to the G594 parent compound GDC-0068, is in phase 1 trials for various tumors. Other AKT inhibitors such as afuresertib and MK-2206 or PI3K inhibitors such as enzastaurin and PX-866 are also being studied.

From these results, we propose that RHOB levels can predict resistance to EGFR-TKI in lung cancer tumors harboring an EGFR mutation. Importantly, based on our results, patients with high RHOB expression levels could be offered an alternative clinical trial

studying the frontline combination of erlotinib with AKT inhibitors. It is possible that RHOB expression levels vary throughout the course of treatment in EGFR-mutated patients. Indeed, the observation that the majority of patients showed an increase in RHOB tumor expression after relapse further supports a role of RHOB in EGFR-TKI resistance.

Overall, the present study offers a new strategy to increase the initial clinical response rate in EGFR-mutated lung cancer patients by providing a molecular rationale for using EGFR-TKI in combination with AKT inhibitor in patients harboring high RHOB tumor levels.

# Materials and Methods

### Cell culture, transfection, and adenoviral transduction and inhibitors

The human NSCLC cell lines HCC4006 (CRL-2871, EGFR del L747-E749, A750P), HCC827 (CRL-2868, EGFR del E749-A750), HCC2935 (CRL-2869, EGFR del E746-T751, S752I), A549 (CCL-185, EGFR WT, KRAS G12S), and H1299 (CRL-5803, EGFR WT, NRAS Q61K) were obtained from the American Type Culture Collection (Manassas, VA, USA). The H3255 cell line (EGFR L858R) was a kind gift from Helene Blons (APHP, Paris, France). All cell lines were cultured in RPMI 1640 medium containing 10% fetal bovine serum (FBS) and were maintained at 37°C in a humidified chamber containing 5% $CO_2$. Cell lines were authenticated and tested for mycoplasma contamination within the experimental time frame.

RNA interference was achieved by transfection of small interfering RNA (siRNA) against RHOB: siB1, 5′-GUCCAAGAAACUGAUG UUA-3′, and siB2, 5′-GCUAAGAUGGUGUUAUUUA-3′, or control (siNeg, SR-CL000-005; Eurogentec) using Lipofectamine® RNAi-MAX™ (Invitrogen Life Science Technologies), following the manufacturer's instructions. Transient transfection with the pCMV6 AKT*myr* plasmid (a kind gift from T. Franke, New York, NY, USA) was performed with JetPRIME according to the manufacturer's instructions. Cells were transduced as described previously with replication-defective (ΔE1, E3) adenoviral vectors expressing RHOB (AdRHOB) or GFP (AdCont) under the transcriptional control of the CMV promoter (Bousquet *et al*, 2009). Erlotinib and G594 were supplied by Roche and Genentech, respectively (South San Francisco, CA, USA).

### Cell proliferation assay

72 h after inhibitor treatment, the relative number of viable cells was measured by incubating cells with the MTS reagent (CellTiter 96® AQueous One Solution Cell Proliferation Assay from Promega), as recommended by the manufacturer. Relative cell survival in the presence of inhibitors was normalized to the untreated cells after background corrections.

### SubG1 analysis

For subG1 analysis, DNA content was assessed by staining ethanol-fixed cells with propidium iodide and monitoring with an Accuri™ C6 flow cytometer (BD Biosciences). The number of

cells with subG1 DNA content was determined with the Accuri C6 software.

### Western blot analysis

Cell extracts were analyzed by Western blotting with primary antibodies against RHOB, ERK1/2 (Santa Cruz Biotechnology), p-ERK (T202/Y204), p-AKT (S473), AKT, pGSK3 (S9), GSK3, p-EGFR (T1173), EGFR, PARP, cleaved caspase-3, caspase-3 (Cell Signaling Technology), or actin (Chemicon). Detection was performed using peroxidase-conjugated secondary antibodies and a chemiluminescence detection kit (Clarity™ ECL; Bio-Rad) with a ChemiDoc™ MP Imaging System (Bio-Rad). Quantifications were carried out for three independent experiments with ImageLab software (Bio-Rad) and normalized to actin.

### Animal studies

Animal experiments were performed with lung-specific tetracycline-inducible human EGFR[L858R] bi-transgenic mice (CCSP-rtTA; tetO-EGFR[L858R]) crossed with *Rhob* null ($Rhob^{-/-}$), heterozygous ($Rhob^{+/-}$), and wild-type ($Rhob^{+/+}$) mice and purified in the 129S2/SvPasCrl strain, as previously described (Calvayrac *et al*, 2014). Approval from the Claudius Regaud Institute Animal Ethics Committee (# ICR-2009-021) was obtained for the use of mice in this animal model and for the study protocols. Animals were housed under controlled temperature and lighting (12/12-h light/dark cycle) and fed with commercial animal feed and water *ad libitum*. All procedures involving animals and their care conformed to institutional guidelines for the use of animals in biomedical research.

Five- to six-week-old mice were fed *ad libitum* with food pellets that contained doxycycline (1 g/kg) for 8 weeks. Then, erlotinib (12.5 mg/kg), G594 (25 mg/kg), or vehicle was injected intraperitoneally daily for 4 days. 24 h after the last injection, the mice were sacrificed by cervical dislocation. The lungs were excised and inflated via intratracheal infusion with 4% buffered formaldehyde and immersion-fixed for 24 h at room temperature before dehydration and paraffin-embedding. Four-micrometer paraffin sections were used for hematoxylin and eosin staining followed by immunohistochemistry using standard procedures. The proliferating index was determined by Ki67 staining (SP6; Thermo Scientific). Transgene expression was evaluated with an anti-EGFR[L858R] antibody (3197; Cell Signaling). Digital slides were blind evaluated by two operators, one of whom was a veterinary pathologist, according to reference papers (Nikitin *et al*, 2004; Politi *et al*, 2006). Detailed methods for tumor grading and quantification of the tumor-to-lung area ratio have been described previously (Calvayrac *et al*, 2014).

### Patients and ethical considerations

We selected patients harboring EGFR-activating mutations (either exon 19 deletion or exon 21 mutation) who had been treated with EGFR-TKI in our institution. Tumor samples were reanalyzed for RHOB expression in our pathology department. All patients had signed an informed consent permitting analyses of tissues. All informed consents were collected and stored in the pathology department. This study was approved by the Ethics of Human

## The paper explained

### Problem

Lung cancer remains the leading cause of cancer-related deaths worldwide. Although impressive treatment advances have been made for patients with non-small-cell lung cancer (NSCLC) whose tumors harbor mutated genes such as EGFR, almost all of them develop resistance mechanisms. To date, no clinically approved biomarker is available to identify the subset of patients that will not benefit from EGFR-tyrosine kinase inhibitor (EGFR-TKI) therapy, and no druggable target has been identified to improve the clinical response rate in resistant patients, highlighting the need for an alternative therapeutic strategy.

### Results

Our findings demonstrate that a high level of the RAS-related GTPase RHOB in the primary lung tumor predicts low progression-free survival in response to EGFR-TKI. Mechanistically, RHOB impairs response to EGFR-TKI through AKT activation. Combining erlotinib with a new-generation AKT inhibitor caused synthetic lethal interaction in EGFR-mutated NSCLC harboring high RHOB levels, suggesting a novel therapeutic strategy to overcome resistance in RHOB-positive patients.

### Impact

The present study offers a new strategy to increase the initial clinical response rate in EGFR-mutated lung cancer patients by providing a molecular rationale for using EGFR-TKI in combination with AKT inhibitor in patients harboring high RHOB tumor levels.

Research Committee at the Pathology Department, Toulouse Hospital, France.

### Immunohistochemistry

Formalin-fixed, paraffin-embedded tissue sections were used for immunohistochemistry (IHC) procedures, as described previously (Calvayrac *et al*, 2014). Briefly, after rehydration, deparaffinized sections were pretreated by microwave epitope retrieval. Endogenous peroxidase activity was quenched and non-specific binding was blocked. For IHC of patient tissues, a RHOB monoclonal antibody was used (C-5, Santa Cruz Biotechnologies, Inc., 1:75). For IHC on mouse lung sections, we used the Ki67 (SP6; Thermo Scientific), ERK1/2 (Santa Cruz Biotechnology), p-ERK (T202/Y204), p-AKT (S473), AKT, and cleaved caspase-3 (Cell Signaling Technology) antibodies, with an Envision kit (DAKO). Sections were lightly counterstained with hematoxylin. Tissues expressing different levels of RHOB were included in each immunohistochemical run to unify any possible discordance in intensity. Two observers (IR, E.C-T), blinded to the patients' status, independently evaluated the extent and intensity of the staining. For RHOB, the intensity of staining was compared with a known external positive control (0: negative; 1+: mild; 2+: moderate; 3+: intense) as previously described (Calvayrac *et al*, 2014), and is shown in Fig 1A. Any discordant independent readings were resolved by simultaneous reviews by both observers.

### Statistics

Continuous variables are presented as their means ($\pm$ standard deviations [SD]) or their medians (with interquartile range [IQR] or range [min–max]), according to their distributions. Categorical variables were summarized by frequency and percentage. The chi-square or Fisher's exact tests were used to compare categorical variables, and Student's *t*-test, variance analysis, or a nonparametric test was used for continuous variables. Paired measurements of continuous data were compared using the Wilcoxon matched-pairs signed-rank test.

All survival times were calculated from the first administration of the TKI drug and estimated by the Kaplan–Meier method with 95% confidence intervals (CIs), using the following first-event definitions: progression or death for progression-free survival (PFS). Surviving patients were censored on the date of the last follow-up. Univariate analysis was performed using the log-rank test.

Pearson correlation analysis was performed to determine the correlation between RHOB levels and erlotinib $IC_{50}$ values.

Tests were two-sided and *P*-values < 0.05 were considered significant. All analyses were conducted using either Stata® version 13.0 (for patient data) or GraphPad Prism 5 software (for *in vitro* and mouse data).

For *in vitro* experiments, data are representative of at least three independent experiments.

**Expanded View** for this article is available online.

## Acknowledgements

This research was financially supported in part by Institut Roche (France), the Institut National de la Santé et de la Recherche Medicale (INSERM), and the Fondation Recherche et Innovation Thérapeutique en Cancérologie, Fondation de France. We thank Bettina Couderc and Catherine Bouchenot for the generation of the adenovirus expressing RHOB. We also thank Anne Casanova for genotyping, Lourdes Gasquet at the Claudius Regaud Institute animal facility, and Helen Blons and Audrey Mansuet-Lupo for kindly providing the H3255 cell line. We also thank the TCGA Research Network (http://cancergenome.nih.gov/) that generated the data used in this study (Cerami *et al*, 2012; Gao *et al*, 2013).

## Author contributions

OC, AS, JMa, AP, and GF contributed to study conception and design and manuscript preparation. OC, JMa, AP, and GF contributed to data analysis and interpretation. OC, IR-L, and EB contributed to development of methodology. OC, CM-D, IR-L, EB, MF, EC-T, IR, NG, SF, JMi, and AL contributed to acquisition of data. AL performed the statistical analysis. EC-T, IR, AL, JC, NM, and SF contributed to administrative, technical, or material support. JMa and GF supervised the study.

## Conflict of interest

The authors declare that they have no conflict of interest.

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
