## [Review Process File · EMBO Molecular Medicine]

The RAS-related GTPase RHOB confers resistance to EGFR-tyrosine kinase inhibitors in non-small cell lung cancer via an AKT-dependent mechanism

Olivier Calvayrac, Julien Mazières, Sarah Figarol, Claire Marty-Detraves, Isabelle Raymond-Letron, Emilie Bousquet, Magali Farella, Estelle Clermont-Taranchon, Julie Milia, Isabelle Rouquette, Nicolas Guibert, Amélie Lusque, Jacques Cadranel, Nathalie Mathiot, Ariel Savina, Anne Pradines, Gilles Favre

Corresponding authors: Gilles Favre and Julien Mazières, INSERM UMR 1037

Review timeline:

Submission date:	03 May 2016
Editorial Decision:	09 May 2016
Author's Appeal:	17 May 2016
Editorial Decision:	20 May 2016
Authors Submit Revision:	23 May 2016
Editorial Decision:	22 June 2016
Revision received:	28 October 2016
Editorial Decision:	09 November 2016
Revision received:	21 November 2016
Accepted:	23 November 2016

Transaction Report:

Editor: Roberto Buccione

1st Editorial Decision

09 May 2016

Thank you for the submission of your manuscript "The RAS-related GTPase RHOB confers resistance to EGFR-tyrosine kinase inhibitors in NSCLC via an AKT-dependent mechanism". I am sorry for the delay in replying due to concomitant editor travel and the increased submissions.

I have now had the opportunity to carefully read your paper and the related literature and I have also discussed it with my colleagues. I am afraid that we concluded that the manuscript is not well suited for publication in EMBO Molecular Medicine and have therefore decided not to proceed with peer review.

You find that in patients with mutated EGFR who had been treated with EGFR-TKI, RHOB tumor tissue levels predict patient response rate to therapy. You also find that, based on the previously described EGFR(L858R)/Rhob(-/-) and EGFR(L858R)/Rhob(+/+) mice, low RHOB expression is associated to a better response to EGFR-TKI. We appreciate that you confirm that RHOB acts via the AKT pathway and that AKT inhibition reverses RHOB-induced resistance to erlotinib.

Although we acknowledge the potential interest of your findings, we find the manuscript better suited to a specialist readership at this stage. We are in fact, also due to the available knowledge and the lack of evidence that RHOB expression is a predictor of response in treatment naïve patients, not persuaded that your manuscript provides the striking level of conceptual advance and translational development we would like to see in an EMBO Molecular Medicine article.

I wish to add however, that considered the potential interest of your findings, I would commit to sending out for peer-review a new manuscript on the same topic if at some time in the near future you have obtained data that would considerably strengthen the clinical relevance of the study along the lines mentioned above.

I am sorry that I could not bring better news.

Author's Appeal

17 May 2016

Thank you for your response. We are very thankful for providing us with thoughtful arguments supporting your decision. I don't want to contest an editorial decision but I would like to clarify one point. You are suggesting below that the major caveat of our study is the lack of clinical relevance. We have the feeling that our clinical data strongly validate the role of RHOB to predict EGFR-TKI response in patients.

In the present study, RhoB expression has been determined in a series of patient including 30 naïve patients that have received EGFR-TKi as first line treatment, 22 patients as second line, and 4 patients as 3rd and 4th line treatment. You will find attached to this email the Kaplan-Meier curves for individual groups (first and 2nd-to-4th line only). As you will see, high RhoB expression in treatment-naïve lung tumor biopsies is strongly associated with a shorter progression free survival to EGFR-TKi. More interestingly RHOB expression in the primary tumor determines the response rate in patients treated with EGFR-TKI in second, third and fourth line of treatment. We can modify the text to emphasize these results.

So we firmly believe that RhoB expression in the primary tumor can be used as a predictive biomarker in this frequent population of patient and can orient the therapeutic strategy.

I hope, that in light with this comment, you would be able to reconsider your final decision.

Thank you again for your consideration.

2nd Editorial Decision

20 May 2016

Thank you for your message asking to reconsider our recent decision on your manuscript entitled "The RAS-related GTPase RHOB confers resistance to EGFR-tyrosine kinase inhibitors in NSCLC via an AKT-dependent mechanism".

We have now re-discussed your manuscript.

We have consequently decided to move forward with peer-review on your manuscript. Before we do so however, I would invite you as you propose yourself, to modify the manuscript to emphasize the demonstration of the direct clinical relevance and translatability of your results (especially including the treatment naïve patients) and including the data you showed us in your letter.

I look forward to reading a new revised version of your manuscript as soon as possible.

Thank you for the submission of your manuscript to EMBO Molecular Medicine. We have now heard back from two Reviewers whom we asked to evaluate your manuscript.

In this case we experienced unusual difficulties in securing three willing and appropriate reviewers. As a further delay cannot be justified I have decided to proceed based on the two available consistent evaluations.

Both Reviewers are generally positive (although Reviewer 1 is more reserved) on the overall relevance and importance of the message conveyed by your manuscript although they express some concerns that require your action. I will not dwell into much detail, as their comments are clear. I would like, however, to highlight a few general points.

Reviewer 1 finds that the clinical data are wanting in both numbers of patients and methodology of analysis. Furthermore, the *in vivo* model protocol is questioned in terms of duration of treatment. The reviewer also mentions other important points including a missing control.

Reviewer 2 also raises two important issues. One is a conflict with earlier data and the other the request to investigate whether indeed RhoB levels increase in EGFR TKI-resistant tumours.

The ensuing cross-commenting discussion also touched upon Reviewer 1's point 1 and highlighted the fact that while indeed a relatively small number of samples was used, it might be challenging to collect more EGFR mutant lung tumor samples and that perhaps IHC might be an acceptable option. Nevertheless, I would encourage you to address these specific concerns including with additional samples as far as realistically possible.

In conclusion, while publication of the paper cannot be considered at this stage, we would be pleased to consider a revised submission, with the understanding that all the Reviewers' concerns must be addressed as outlined above.

Please note that it is EMBO Molecular Medicine policy to allow a single round of revision only and that, therefore, acceptance or rejection of the manuscript will depend on the completeness of your responses and on the outcome of the required experimentation included in the next, final version of the manuscript.

As you might know, EMBO Molecular Medicine has a "scooping protection" policy, whereby similar findings that are published by others during review or revision are not a criterion for rejection. However, I do ask you to get in touch with us after three months if you have not completed your revision, to update us on the status. Please also contact us as soon as possible if similar work is published elsewhere.

Please note that EMBO Molecular Medicine now requires a complete author checklist (<http://embomolmed.embopress.org/authorguide#editorial3>) to be submitted with all revised manuscripts. Provision of the author checklist is mandatory at revision stage; The checklist is designed to enhance and standardize reporting of key information in research papers and to support reanalysis and repetition of experiments by the community. The list covers key information for figure panels and captions and focuses on statistics, the reporting of reagents, animal models and human subject-derived data, as well as guidance to optimise data accessibility.

I also suggest that you carefully adhere to our guidelines for publication in your next version, including presentation of statistical analyses and our new requirements for supplemental data (see also below) to speed up the pre-acceptance process in case of a favourable outcome.

I look forward to seeing a revised form of your manuscript as soon as possible.

***** Reviewer's comments *****

Referee #1 (Comments on Novelty/Model System):

The question addressed, namely molecular mechanisms of resistance to EGFR inhibitors is attractive and the animal models used are suitable.

Referee #1 (Remarks):

Manuscript Number: EMM-2016-06646

Oliver Calvayrac et al.: The RAS-related GTPase RHOB confers resistance to EGFR-tyrosine kinase inhibitors in NSCLC via an AKT-dependent mechanism

Corresponding Author: Dr. Gilles Favre

Soon after the approval of the first anti-cancer, molecular targeted drugs, the frequent emergence of patient resistance has become a topic of major interest to both medical and molecular oncologists. The authors of EMM-2016-06646 address lung cancer resistance to erlotinib, an EGFR-specific kinase inhibitor. By analyzing a group of erlotinib-naïve patients, they found that RHOB levels significantly associated with time to disease progression in a group of lung cancer patients. Based upon this observation, they surveyed a small set of cancer cell lines and reached a similar conclusion. More importantly, their experiments using transgenic mice inducibly expressing an oncogenic mutant of EGFR, which they crossed with RHOB-defective mice, provided strong support to their working hypothesis. This animal model was employed to test another hypothesis, namely: RHOB activates AKT, thereby confers resistance to erlotinib. Indeed, when treating cells with erlotinib the observed inhibition of AKT, but this was abolished in RHOB-overexpressing cells. Likewise, treatment of mice with a combination of inhibitors, to AKT and to EGFR, more strongly inhibited their tumours and induced apoptosis than each drug alone.

The manuscript is well written and appropriately combines in vitro studies with animal models and patient data. However, several issues must be addressed in order to strengthen the conclusions or increase clarity.

1. Figure 1 presents clinical data from 56 lung cancer patients, who were stratified into 4 groups according to RHOB abundance. The analysis was performed using immunohistochemistry (IHC). Both the number of patients and the use of IHC are problematic. In order for RHOB levels to become a useful and reliable biomarker, more specimens and a less variable methodology are needed. The authors might address involvement of microRNAs and copy number aberrations in the variable abundance of RHOB. Alternatively, they might use PCR or the promoter of RHOB, fused to a reporter gene, to uncover potential determinants of RHOB expression.
2. Do RHOB levels predict prognosis of lung cancer patients who present no EGFR mutations? Could it be that RHOB overexpression is the result of strongly oncogenic mutations like p53 or KRAS, hence associate with lack of response to erlotinib and other drugs?
3. Figure 2 presents results obtained using a transgenic animal model. Why treatment with erlotinib was limited to 4 days? Do the differences among mice disappear following a longer treatment? Note that RHOB homo- and heterozygous show minimal differences, why? Explain why wild type mice exhibited relatively small tumours and lower Ki67 signals.
4. Figure 4 presents experiments performed with four cell lines expressing mutants of EGFR. It would be worthwhile adding a cell line expressing a wild type form of EGFR.
5. Previous studies implicated pathways other than AKT in the action of EGFR mutations. They include STAT and ERK, along with other signaling pathways. The corresponding references should be added.

Referee #2 (Comments on Novelty/Model System):

Methods are adequate throughout. They include tumor tissue interrogation, in vitro experiments and in vivo work using transgenic mouse model of EGFRmutant lung adenocarcinoma. Identification of RHOB in resistance to EGFR TKI could have implications in the clinics, as biomarker of sensitivity or resistance. Additionally, AKT inhibition may be used in combination with EGFR TKI like demonstrated in this study.

Referee #2 (Remarks):

In their manuscript, Calvayrac et al. describe an important role for RHOB small GTPase in the resistance of EGFR-mutant lung tumors to EGFR tyrosine kinase inhibitors (TKI), which are commonly used treatments in the lung cancer clinics. They provide compelling evidence for the importance of RHOB including expression correlation studies from lung tumor tissues, genetic manipulation in vitro and use of Rhob deficient mice crossed to a mutant Egfr transgenic mouse model of lung adenocarcinoma. In vitro, RHOB knockdown decreases, while RHOB over-expression increases IC50 of erlotinib. They provide scientific demonstration that RHOB enables sustained AKT activity and hence resistance. In Rhob WT or KO tumors, pERK decreases dramatically upon erlotinib treatment, but there is decrease of pAkt only in Rhob KO tumors. Constitutively active AKT expressed in vitro decreases efficacy of erlotinib phenocopying Rhob over-expression. Finally an AKT small molecule inhibitory compound re-sensitizes RHOB over-expressing EGFR mutant lung tumor cells to erlotinib treatment, in vitro, and in vivo in Rhob WT mice. These are very important findings with potential immediate implications in the clinics, and molecular mechanisms are clearly provided. Manuscript is well written. I have a few minor comments:

- 1) Text, page 3, line 4, I guess the authors wanted to write "tumor suppressor" instead of "gene suppressor".
- 2) It is surprising to see an apparent absence of effect of erlotinib in the Egfrmutant, Rhob +/- transgenic mice (Fig. 6A, C, B and D blue compared to green dots), when compared to the initial characterization of this mouse model (Politi 2006), where tumors disappeared under erlotinib treatment. I believe Politi et al had used mice whose tumors express Rhob. Hence, the authors should discuss the contrasting results. Are there clear reasons that can explain the discrepancies that readers would need to know?
- 3) Although the authors already mention this point at the end of their discussion (page 11), one is tempted to speculate that RHOB expression levels increase in EGFR TKI resistant tumors. Do the authors have access to some tumor tissue before treatment and re-biopsied upon resistance, to begin to test this hypothesis?

2nd Revision – author's response

09 November 2016

Referee #1 (Remarks):

We want to thank reviewer 1 for his thoughtful comments that have helped us to greatly improve the quality of our manuscript.

1. a. Figure 1 presents clinical data from 56 lung cancer patients, who were stratified into 4 groups according to RHOB abundance. The analysis was performed using immunohistochemistry (IHC). Both the number of patients and the use of IHC are problematic. In order for RHOB levels to become a useful and reliable biomarker, more specimens and a less variable methodology are needed.

Answer to these comments will be detailed in 2 parts:

a) We fully agree with Reviewer 1 that the inclusion of more patients in the study would indubitably strengthen the clinical relevance of our results. In the allotted time for revision, we included 43 new EGFR-mutated lung adenocarcinoma patients of which 18 were treated with Gefitinib, 20 with Erlotinib and 2 patients with Afatinib, in first line (n=33), second line (n=6) or third line (n=1)

setting. 3 patients didn't receive EGFR-TKI and were excluded from the PFS study, although these 3 samples were kept in the series that compared protein and mRNA RhoB levels as described below (Graph 1).

Graph 1. Repartition of samples for the new series of EGFR-mutated lung adenocarcinoma patients.

Median progression-free survival (PFS) was 10.35 months [6.21; 14.09] for this new series of 40 patients (Graph 2).

Graph 2. Progression-free survival of EGFR-TKI-treated patients with EGFR-mutated lung tumors (blue curve: first series, green curve: second series, red curve: pooled series).

We performed RHOB immunohistochemistry on FFPE tumor samples collected at the time of diagnosis using the same protocol as described in the first series. Two observers (IR, E.C-T), blinded to the patients' status, independently evaluated the extent and intensity of the staining. Samples from the first series that corresponded to the 4 previously defined RhoB levels (null: 0, weak: +, moderate: ++ and high: +++) were also included as internal controls. Samples with null or weak staining were considered as low RHOB patients and samples with moderate or high staining as high RHOB patients. Data obtained with this new series confirm that high RHOB levels induce EGFR-TKI resistance in EGFR-mutated patients, as median PFS was 15.9 months (CI95% [12.2; 18.3]) for patients with low RHOB expression (0 and +) and only 8.1 months (CI95% [3.8 ; 9.7]) for patients with high RHOB expression (++ and +++) ($p=0.0004$, Graph 3B). Given that both series were collected and processed in the same institute and had similar characteristics (see comparative analysis below in table 1), we combined both series of patient to obtain a new series of 96 patients that we included in the article in replacement of the first series of 56 patients (Graph 3C below and Figure 1 in the article).

Graph 3. Progression-free survival of EGFR-TKI-treated patients with EGFR-mutated lung tumors, according to RHOB expression, assessed by immunohistochemistry (low RHOB group= negative + weak staining, blue curves; high RHOB group = moderate + high staining, red curves) for the first series (A), the second series (B) and pooled series (C). D. Progression-free survival of patients who received EGFR-TKI as first-line therapy (n=63). E. Progression-free survival of patients who received EGFR-TKI as 2nd-line (n=28), 3rd-line (n=3) or 4th-line (n=2) therapy.

Characteristics of the pooled series are presented in the article as new Appendix Table S1. Median progression-free survival (PFS) was 12.06 months (CI95% [8.11; 13.99]) for the whole population of 96 patients (in article: new Fig EV1A), which is almost identical to the first series of 56 patients. In the pooled series, PFS was still not statistically different between the RHOB (0) and RHOB (+) groups and between the RHOB (++) and RHOB (+++) groups (in article: new Fig EV1B), allowing us to group RHOB (0) with (+) and RHOB (++) with (+++). Median PFS was 15.3 months (CI95% [13.1; 18.2]) for patients with low RHOB expression (0 and +) and 5.6 months (CI95% [3.6; 6.4]) for patients with high RHOB expression (++ and +++) ($p < 0.0001$; Graph 3C, in article: new Figure 1D). Univariate analysis showed that TKI type was significantly associated to PFS but not age, sex, number of previous lines of chemotherapy or type of EGFR mutation (in article: new Appendix Table S1). We performed a multivariate analysis to demonstrate that RHOB is an independent prognostic factor of PFS ($p < 0.001$; in article: Appendix Table S2). We then analyzed RHOB expression on tumor biopsies of patients receiving EGFR-TKI as a first line (Graph 3D, in article: new Figure 1E) or as second-to-fourth lines (Graph 3E, in article: new Figure 1F) therapy. Importantly, high RHOB expression was still strongly associated with a shorter PFS to EGFR-TKI, regardless the line of EGFR-TKI therapy, indicating that the predictive value of RHOB was not affected by chemotherapeutic treatments.

	Pool n=96	Series 1 n=56	Series 2 n=40	p value
Age at initiation of TKi treatment				p= 0.5276
Median	66.0	65	67.5	
(Range)	(37.0 : 90.0)	(37.0 : 90.0)	(44.0 : 81.0)	
Age at initiation of TKi treatment				p= 0.5120
≤ 65 years	47 (49.0%)	29 (51.8%)	18 (45.0%)	
> 65 years	49 (51.0%)	27 (48.2%)	22 (55.0%)	
RHOB staining				p= 0.1768
0	18 (18.8%)	10 (17.9%)	8 (20.0%)	
+	39 (40.6%)	27 (48.2%)	12 (30.0%)	
++	30 (31.3%)	13 (23.2%)	17 (42.5%)	
+++	9 (9.4%)	6 (10.7%)	3 (7.5%)	
Stade				p= 1.0000
IIIb	4 (4.2%)	2 (3.6%)	2 (5.0%)	
IV	92 (95.8%)	54 (96.4%)	38 (95.0%)	
Type of EGFR-TKI				p= 0.1116
Gefitinib	51 (53.1%)	33 (58.9%)	18 (45.0%)	
Erlotinib	43 (44.8%)	23 (41.1%)	20 (50.0%)	
Afatinib	2 (2.1%)	0 (0.0%)	2 (5.0%)	
Type of EGFR-TKI				p= 0.1776
Gefitinib	51 (53.1%)	33 (58.9%)	18 (45.0%)	
Erlotinib or Afatinib	45 (46.9%)	23 (41.1%)	22 (55.0%)	
Time between diagnosis and beginning of EGFR-TKI treatment				p= 0.9097
Median	1.9	1.9	2.0	
(Range)	(0.1 : 36.8)	(0.2 : 25.4)	(0.1 : 36.8)	
Time between diagnosis and beginning of EGFR-TKI treatment				p= 0.4805
≤ 3 months	60 (63.2%)	37 (66.1%)	23 (59.0%)	
> 3 months	35 (36.8%)	19 (33.9%)	16 (41.0%)	
Progression-Free Survival				p= 0.4653
Median	12.06	12.06	10.35	
[IC95%]	[8.11 ; 13.99]	[6.90 ; 14.92]	[6.21 ; 14.09]	

Table 1. Comparative analysis between the first and the second series of patients.

b) To answer to the requirement of a less variable methodology than IHC, wisely proposed by Reviewer 1, we analyzed RHOB levels by quantitative real-time PCR. From the new series of patients, tumor material was still available for 29 samples. Tumor areas (predetermined by an anatomopathologist) from 4-to-6 freshly cut sections of FFPE tissue (10 μm thick) were collected using a scalpel, RNA extraction was performed using the miRNeasy FFPE (Qiagen) that allow extraction of total RNA including miRNA, following manufacturer's instructions. After extraction, 4 out of the 29 samples (13.8%) didn't show quantifiable amount of RNA (determined by NanoDrop® ND-1000 Spectrophotometer). For the remaining 25 samples, 100 ng of total RNA was reverse-transcribed using the iScript cDNA synthesis kit, following the manufacturer's instructions (Bio-Rad Laboratories), and quantitative real-time PCR of RhoB, β-actin and GAPDH mRNA was performed with a CFX96 detection system (Bio-Rad), using iQ SYBR Green Supermix (Bio-Rad) and the sequence-specific primers for RhoB (forward, 5'- TTGTGCCTGTCCTAGAAGTG-3'; reverse, 5'- CAAGTGTGGTCAGAATGCTAC-3'), β-actin (forward, 5-

TCCTGGAGAAGAGCTACGA -3'; reverse, 5- AGGAAGGAAGGCTGGAAGAG-3') and GAPDH (forward, 5- TGCACCACCAACTGCTTAGC -3'; reverse, 5- GGCATGGACTGTGGTCATGAG-3'). The relative RhoB mRNA expression was calculated according to the $2^{-\Delta\Delta Cq}$ method, normalized to the β -actin and GAPDH mRNA levels. Two samples showed very late or no amplification of any of the three genes, probably due to a high degree of RNA degradation, and were excluded from the analysis.

As shown in Graph 4A and 4B, we observed a significant correlation between RHOB mRNA expression (determined by RT-qPCR) and RHOB protein staining (determined by IHC) in the 23 tumor samples. Concordantly, a significant correlation between RHOB mRNA and protein levels was also observed in the EGFR-mutated cell lines (Graph 4C).

Graph 4. Correlation between RHOB mRNA and protein levels in patients and in EGFR-mutated cell lines. A) RHOB mRNA levels in tumors according to the corresponding IHC score (null: 0, weak: +, moderate: ++ and high: +++), including correlation analysis (in red, right Y axis). B) RHOB mRNA levels in the RHOB-low (0/+) and the RHOB-high (++/+++) groups. C) Correlation between RHOB mRNA and protein levels in four EGFR-mutated cell lines, determined by RT-qPCR and Western Blot, respectively.

We next aimed to determine whether RHOB mRNA levels were predictive of the response to EGFR-TKI. PFS analysis was performed on the 21 patients who received EGFR-TKI treatment (2 patients didn't and were excluded from this analysis as mentioned before, see Figure 1). As shown in Graph 5A, median PFS was 14.0 months (CI95% [5.7; 27.1]) for patients with low RHOB mRNA expression (mRNA expression below median) and 9.0 months (CI95% [1.6; 13.7]) for patients with high RHOB mRNA expression (mRNA expression above median) ($p=0.0643$). PFS analysis using corresponding RHOB IHC showed similar results, as median PFS was 9.0 months (CI95% [5.8 ; 13.7]) for patients with low RHOB (0 and +) and 20.6 months (CI95% [12.3 ; 27.2]) for patients with high RHOB (++ and +++) ($p=0.0059$; Graph 5B). Positive correlation between RHOB mRNA and IHC was still significant in this subgroup of 21 patients (Graph 5C). These two experiments didn't reach the same level of significance, which may be explained by the small series of patients, the low number of remaining samples for the analysis, a high degree of RNA degradation (some FFPE samples are several years old) or the fact that macrodissection performed to isolate RNA from tumor tissue may have been contaminated by non-tumoral adjacent tissue and thus may have "diluted" the information of RHOB mRNA expression in tumor.

Altogether, these results suggest that both IHC and mRNA may be adapted to evaluate RHOB expression in tumor sample. Results of the correlation analysis between RHOB mRNA and protein levels as well as PFS analysis according to RHOB mRNA expression have been included in the article as new Appendix Fig S1.

Graph 5. Progression-free survival of EGFR-TKI-treated patients with EGFR-mutated lung tumors, according to RHOB expression assessed by quantitative real-time PCR (low RHOB group= under median, blue curve; high RHOB group = above median, red curve) for the second series of 21 patients (A), or assessed by IHC in the corresponding samples (low RHOB group= 0/+ staining; high RHOB group = ++/+++ staining). C) RHOB mRNA levels in the RHOB-low (0/+) and the RHOB-high (++) groups.

1. b. The authors might address involvement of microRNAs and copy number aberrations in the variable abundance of RHOB. Alternatively, they might use PCR or the promoter of RHOB, fused to a reporter gene, to uncover potential determinants of RHOB expression.

As proposed by Reviewer 1, we aimed to address involvement of microRNAs and copy number aberrations in the variable abundance of RHOB.

a. microRNA: It was previously described that RHOB expression is regulated by several miRNA including miR-19a/b (Glorian *et al*, 2011; Tan *et al*, 2015), miR-21 (Connolly *et al*, 2010; Liu *et al*, 2011; Shi *et al*, 2013; Yang *et al*, 2013), miR-223 (Sun *et al*, 2010; Zeng *et al*, 2016). Moreover RhoB mRNA possesses a predicted target sequence for miR-30 (especially miR-30b and miR-30e, as determined in silico using the microrna.org online tool). From the 23 RNA samples used for RHOB mRNA quantification, RNA was still available for 20 samples (see Figure 1). 100 ng of total RNA (including miRNA) was reverse-transcribed using the Universal cDNA Synthesis Kit II, following the manufacturer's instructions (Exiqon), and quantitative real-time PCR of miR-19a, miR-19b, miR-21, miR-30b, miR-30e and miR-223 was performed with a CFX96 detection system (Bio-Rad), using miRCURY LNA microRNA PCR ExiLent SYBR® Green (Exiqon) and the LNA™ PCR primer set for hsa-miR-19a-3p (#205862), hsa-miR-19b-3p (#204450), hsa-miR-21 (#204230), hsa-miR-30b-5p (#204765), hsa-miR-30e-3p (#204410), hsa-miR-223-3p (#204256), and reference primer mix for SNORD38B (#203901) and SNORD49A (#203904) for normalization. As shown in Table 2, a significant negative correlation was observed between miR-21 expression and RHOB levels (determined by either IHC or by RT-qPCR), and a negative correlation was also observed with miR-19a and miR-19b although significance was not reached.

RHOB expression determined by IHC (n=20)					RHOB expression determined by RT-qPCR (n=20)	
		0/+ (n=7)	++/+++ (n=13)	p value	correlation coefficient	p value
miR-19a	Median (range)	1.00 (0.13 : 3.74)	0.45 (0.10 : 2.23)	0.1655	-0.3248	0.1623
miR-19b	Median (range)	2.23 (0.17 : 5.14)	0.88 (0.18 : 3.73)	0.4054	-0.2241	0.3423
miR-21	Median (range)	1.01 (0.57 : 3.70)	0.36 (0.09 : 2.46)	0.0079	-0.6150	0.0039
miR-30b	Median (range)	1.00 (0.50 : 8.55)	1.80 (0.36 : 15.49)	0.5523	0.2872	0.2195
miR-30e	Median (range)	3.49 (1.00 : 8.23)	2.66 (1.48 : 14.96)	0.6630	0.0647	0.7865
miR-223	Median (range)	1.00 (0.43 : 6.80)	1.31 (0.25 : 19.63)	0.5006	0.2271	0.3357

Table 2. Correlation between expression of several miRNA and RHOB levels determined by RT-qPCR or IHC staining in tumors of patients harboring EGFR-activating mutations.

b. RHOB Copy Number Aberration: In order to assess whether copy number variation could be a determinant of RHOB expression, we used The Cancer Genome Atlas (TCGA) lung adenocarcinoma database that gives information on mRNA levels, putative copy-number alterations and mutations for the entire genome of 512 tumor samples, from which 58 had information for corresponding normal adjacent tissue. As represented in Graph 6A, 327 tumors (63.9%) had normal ploidy for RHOB, 34 (6.6%) displayed RHOB heterozygous loss, 148 (28.9%) presented a gain and 3 (0.6%) had an amplification on RHOB locus. Worthy of note, neither mutation nor deletion were found on RHOB gene, consistent with previous observations in lung cancer (Sato *et al*, 2007) and in our laboratory (unpublished data) or in other cancers like head and neck carcinoma (Adnane *et al*, 2002) or breast cancer (Fritz *et al*, 2002). Repartition of RHOB putative copy number alteration was similar for the subgroup of 33 lung tumors harboring an EGFR mutation (normal ploidy: 57.6%; heterozygous loss: 9.1%; gain: 33.3%; amplification: 0%). RHOB mRNA levels were significantly lower in lung tumors when compared to normal tissue (Graph 6B), which confirm previous observations reported by us and others (Mazieres *et al*, 2004; Sato *et al*, 2007). Importantly, RHOB mRNA expression in tumors was not significantly affected by copy number alterations, as RHOB expression was not significantly different in the “RHOB heterozygous loss” group or the “RHOB gain” group when compared to the “RHOB normal diploid” group (Graph 6B). This was the case for both the entire cohort of 512 patients and for the 33 EGFR mutated subgroup. These observations are in perfect concordance with Sato *et al*.’s work (Sato *et al*, 2007) in which they also observed a significant downregulation of RHOB in NSCLC but they “*did not find a significant inverse correlation between the LOH and IHC data or between the LOH and RT-PCR data*”. Altogether, these data suggest that differences of RHOB expression observed in tumors didn’t seem to be caused by a variation of its copy number.

Graph 6. A) Percentage of RHOB mutation and putative copy number alterations in tumor samples from 512 lung adenocarcinoma patients, or in selected patients harboring an EGFR-mutation (n=33). Putative copy number was determined using GISTIC 2.0 (values: -2: homozygous deletion, -1: heterozygous loss, 0: diploid, +1: Gain, +2: amplification). B) Correlation between RHOB mRNA expression (Y axis, log₂ scale) and copy number (X axis). All data were retrieved from the lung adenocarcinoma TCGA database, using the cBioPortal for Cancer Genomics tool (cbioportal.org).

c. Other determinants of RHOB expression: Several studies indicate that RHOB expression can be repressed by histone acetylation at its promoter region, and that treatment with histone deacetylase inhibitors (HDACi) can reverse this downregulation (Delarue *et al*, 2007; Mazieres *et al*, 2007; Sato *et al*, 2007; Wang *et al*, 2003). In order to validate these observations in EGFR-mutated tumor cells and provide a possible mechanism of the variable abundance of RHOB, we treated the four cell lines used in present study (HCC827, HCC4006, H3255 and HCC2935) with the HDACi Trichostatin A (TSA) for 24h at 1 μ M. As shown in Graph 7, TSA treatment induced RHOB expression in the four cell lines, with an inversely proportional amplitude to the basal RHOB level, supporting a role of histone acetylation in RHOB downregulation.

Graph 7. HDAC inhibition induces RHOB overexpression in EGFR-mutated cell lines. HCC827, HCC4006, H3255 and HCC2935 cell lines were treated for 24h with Trichostatin A (TSA) at 1 μ M and RHOB expression was assessed by Western Blot and normalized by actin.

Altogether these data suggest the RHOB expression in lung cancer cells depends on epigenetic mechanism involving particularly miR-21 expression, a well-known oncomir, and chromatin acetylation. This information has been featured as new Appendix Table S3, new Appendix Figure S6 and new Appendix Figure S7, and is discussed in the main text of the article.

2. Do RHOB levels predict prognosis of lung cancer patients who present no EGFR mutations? Could it be that RHOB overexpression is the result of strongly oncogenic mutations like p53, KRAS, hence associate with lack of response to erlotinib and other drugs?

In the present study, we didn't evaluate prognostic values of RhoB in EGFR WT lung cancer as we focused on EGFR-mutated tumors and the role of RHOB in the response to EGFR-TKI treatment. However, the prognosis value of RHOB in EGFR WT lung tumors has been recently evaluated in one of our studies (Calvayrac *et al*, 2014), using a cohort of patients from the IFCT-0401 clinical trial (Cadranel *et al*, 2009) wherein the purpose was to evaluate the efficacy and safety of gefitinib as a first-line treatment for patients with adenocarcinoma with lepidic subtype (formerly known as bronchioloalveolar carcinoma). In this series of patients, low RHOB expression was associated with shorter survival time as median overall survival (OS) for the low RHOB group was 13.2 month (IC95% [7.7; 19.6]) whereas median OS was not reached in patients with high RhoB expression ($p = 0.009$; see Figure 2B in (Calvayrac *et al*, 2014)), which confirms that RHOB is a potential prognostic biomarker for this particular subtype of lung cancer. However, no difference of PFS was found between the low RHOB group (PFS= 3.3 month; IC95% [2.0; 9.5]) and the high RHOB group (PFS= 3.2 month; IC95% [2.0; 11.3]). These results are consistent with the fact that lung adenocarcinoma patients with no EGFR mutation do not benefit from EGFR-TKi treatment (Herbst *et al*, 2005), and explain why RHOB cannot predict the response to EGFR-TKI treatment in these patients.

To answer the second part of the question, we analyzed a potential effect of the major oncogenic mutations found in EGFR-mutated lung adenocarcinoma on RHOB expression using the TCGA database. As shown in Graph 8A, p53 mutations were found in 20/33 EGFR-mutated tumors (60%), KRAS mutations in 1/33 (3%), BRAF mutations in 4/33 (12%) and PIK3CA mutations in 1/33 (3%). None of the mutations were associated with RHOB mRNA expression, although a higher frequency of p53 mutations seemed to be associated with lower levels of RHOB without reaching significance (Graph 8B). This information has been added to the article as new Appendix Fig S6C-D.

These results strongly suggest that RHOB is not associated with relevant oncogenic mutations and is an independent marker of the response to EGFR inhibitors in EGFR-mutated patients. Unfortunately, information for others drugs are not available.

Graph 8. A. Correlation between RHOB expression and the major oncogenic mutations found in EGFR-mutated lung adenocarcinomas, based on the TCGA database. B. Expression of RHOB mRNA in p53 WT and p53 mutated lung adenocarcinomas, based on the TCGA database.

3. Figure 2 presents results obtained using a transgenic animal model. Why treatment with erlotinib was limited to 4 days? Do the differences among mice disappear following a longer treatment? Note that RHOB homo- and heterozygous show minimal differences, why? Explain why wild type mice exhibited relatively small tumours and lower Ki67 signals.

Longer erlotinib treatments were not performed in this study, although we agree with Reviewer 1 that longer treatments could potentially reveal more information on the kinetics of response between *Rhob* WT and *Rhob* KO mice. However, in the initially described mouse model, a complete response seems to be observable after a 4 day treatment in responding mice (treated with erlotinib at 25 or 50 mg/Kg/day), and longer treatments did not seem to provide additional benefits, although these benefits were clearly observed between 2 days and 4 days of treatment (see table 1 in (Politi *et al*, 2006)). In our *in vivo* study, the purpose was to evaluate the role of *Rhob* loss on erlotinib sensitivity but also to determine a potential synergic effect with the combination of AKTi and EGFR-TKi treatments, which indeed could be achieved with a 4 days treatment.

The absence of difference between RHOB homo- and heterozygous mice has already been observed in a previous study using this mouse model (Calvayrac *et al*, 2014). Indeed, we observed that decrease of RHOB expression *in vivo*, even partially (in *Rhob* heterozygous mice), induced a significant increase in both number and size of tumor nodules and was also correlated with a greater proliferating index (determined by Ki67 staining). These results prompted us to propose RHOB as a potential haplo-insufficient tumor suppressor gene that reflects the partial diminution of RhoB protein levels observed during lung tumor progression (Mazieres *et al*, 2004).

4. Figure 4 presents experiments performed with four cell lines expressing mutants of EGFR. It would be worthwhile adding a cell line expressing a wild type form of EGFR.

As proposed by Reviewer 1, we performed control experiments with two cell lines expressing a wild type form of EGFR (A549 and H1299). As expected, both cell lines were resistant to Erlotinib (Graph 9A) and neither AKT nor ERK pathways were affected by treatment, probably due to the absence of activated EGFR (Graph 9B). RHOB overexpression had no effect on cell viability or signaling pathways (Graph 9A and 9B). These results have been included as new Figure EV3.

Graph 9. RHOB overexpression does not affect response to erlotinib in EGFR WT cell lines. **A.** A549 or H1299 cells were transduced with control (AdCont) or RHOB-overexpressing adenoviruses (AdRHOB) and treated with increasing doses of erlotinib. The surviving cell fraction was determined by an MTS assay after 72 h and compared to untreated cells. **B.** A549 and H1299 cells were transduced with control (AdCont) or RHOB-overexpressing (AdRHOB) adenoviruses and treated for four hours with erlotinib at 1 μ M. The phosphorylation status of AKT, ERK1/2 and EGFR was assessed by western blotting and normalized according to total protein levels. RHOB overexpression was also monitored by western blotting. EGFR-mutated HCC4006 cells were used to monitor Erlotinib efficiency (right panel).

5. Previous studies implicated pathways other than AKT in the action of EGFR mutations. They include STAT and ERK, along with other signaling pathways. The corresponding references should be added.

Following sentence has been added to the introduction:

Resistance mechanisms to EGFR-TKi could be mediated mediated by a bypass reactivation of one or several key proliferation and survival signaling pathways downstream to EGFR, mainly PI3K (phosphatidylinositol 3-kinase)/AKT (Engelman et al, 2007), MEK (mitogen-activated protein kinase kinase)/ERK (extracellular signal-regulated kinase) (Ercan et al, 2012), or STAT (signal transducer and activator of transcription) pathways (Lee et al, 2014), among others."

Referee #2 (Remarks):

We thank reviewer 2 for its very positive comments concerning our manuscript.

Minor comments:

1) Text, page 3, line 4, I guess the authors wanted to write "tumor suppressor" instead of "gene suppressor".

This has been changed in the revised version of the manuscript.

2) *It is surprising to see an apparent absence of effect of erlotinib in the Egfrmutant, Rhob +/- transgenic mice (Fig. 6A, C, B and D blue compared to green dots), when compared to the initial characterization of this mouse model (Politi 2006), where tumors disappeared under erlotinib treatment. I believe Politi et al had used mice whose tumors express Rhob. Hence, the authors should discuss the contrasting results. Are there clear reasons that can explain the discrepancies that readers would need to know?*

We thank Reviewer 2 for this pertinent comment that, albeit needs clarification. It's true that in the initial characterization of this mouse model, Politi *et al.* (Politi *et al.*, 2006) used RHOB WT mice that do express RHOB. However, to obtain an objective response to erlotinib treatment, Politi *et al.* needed to treat mice with a high dose of the compound (>25 mg/Kg/day, see table 1 of (Politi *et al.*, 2006)) which is much more elevated than the dose used in clinic (standard dosage for patients is 150 mg/day, that would correspond to 2 mg/Kg/day for a patient weighing 75 Kg). Based on Politi's data, we thus decided to use a lower dosage that didn't show efficiency (12.5mg/Kg/day) in order to evaluate whether RHOB loss could sensitize tumors to erlotinib treatment, and indeed that's what happened.

In order to clarify this point, we modified the corresponding sentence in the result section of the manuscript:

“To evaluate the effect of *Rhob* loss on erlotinib sensitivity *in vivo*, we performed a four-day treatment with erlotinib at 12.5 mg/kg/d that did not show objective response in the initially described mouse model (Politi *et al*, 2006). In these conditions, erlotinib treatment induced a strong anti-tumoral response in *EGFRL858R/Rhob*^{-/-} and *EGFRL858R/Rhob*^{+/-} mice whereas no significant tumor shrinkage was observed in *EGFRL858R/Rhob*^{+/+} mice (Fig 2A).”

3) Although the authors already mention this point at the end of their discussion (page 11), one is tempted to speculate that *RHOB* expression levels increase in EGFR TKI resistant tumors. Do the authors have access to some tumor tissue before treatment and re-biopsied upon resistance, to begin to test this hypothesis?

Again, we thank Reviewer 2 for the pertinence of his question that prompted us to deepen our knowledge on the role of *RHOB* in erlotinib resistance. We had access to 11 re-biopsies after EGFR-TKI relapse and corresponding tumor tissue before treatment (diagnostic biopsies), thus we performed *RHOB* immunostaining on the 22 samples in parallel using the same protocol as described earlier in order to determine the evolution of *RHOB* levels before and after EGFR-TKI relapse. Of interest, 4 of the 11 diagnostic biopsies came from the first series of 56 patients initially described in the first version of the article, and re-staining of these 4 samples gave the same *RHOB* staining score as the first reading, which confirms the reliability and robustness of the method used. As shown in Graph 10, 8/11 tumors showed an increase in *RHOB* expression (of which 5 tumors had a one-score increase and 3 tumors had a two-score increase), 1/11 tumor had no change in *RHOB* levels and 2/11 tumors had a one-score decrease (++) to (+). These important new results, although obtained on a reduced number of patients, indicate that EGFR-TKI relapse is often accompanied by an overexpression of *RHOB*, which supports a role of this GTPase in the resistance to EGFR-TKI.

Graph 10. *RHOB* immunostaining score in EGFR-mutated lung tumors before treatment and after EGFR-TKi relapse.

These results have been included in the revised version of the manuscript (new Figure 1G).

References

Adnane J, Muro-Cacho C, Mathews L, Sebti SM, Munoz-Antonia T (2002) Suppression of rho B expression in invasive carcinoma from head and neck cancer patients. *Clinical cancer research : an official journal of the American Association for Cancer Research* 8: 2225-2232

Cadranel J, Quoix E, Baudrin L, Mourlanette P, Moro-Sibilot D, Morere JF, Souquet PJ, Soria JC, Morin F, Milleron B *et al* (2009) IFCT-0401 Trial: a phase II study of gefitinib administered as first-line treatment in advanced adenocarcinoma with bronchioloalveolar carcinoma subtype. *J Thorac Oncol* 4: 1126-1135

Calvayrac O, Pradines A, Raymond-Letron I, Rouquette I, Bousquet E, Lauwers-Cances V, Filleron T, Cadranel J, Beau-Faller M, Casanova A *et al* (2014) *Rhob* determines tumor aggressiveness in a murine *EGFRL858R*-induced adenocarcinoma model and is a potential prognostic biomarker for Lepidic lung cancer. *Clinical cancer research : an official journal of the American Association for Cancer Research* 20: 6541-6550

- Connolly EC, Van Doorslaer K, Rogler LE, Rogler CE (2010) Overexpression of miR-21 promotes an in vitro metastatic phenotype by targeting the tumor suppressor RHOB. *Mol Cancer Res* 8: 691-700
- Delarue FL, Adnane J, Joshi B, Blaskovich MA, Wang DA, Hawker J, Bizouarn F, Ohkanda J, Zhu K, Hamilton AD *et al* (2007) Farnesyltransferase and geranylgeranyltransferase I inhibitors upregulate RhoB expression by HDAC1 dissociation, HAT association and histone acetylation of the RhoB promoter. *Oncogene* 26: 633-640
- Engelman JA, Zejnullahu K, Mitsudomi T, Song Y, Hyland C, Park JO, Lindeman N, Gale CM, Zhao X, Christensen J *et al* (2007) MET amplification leads to gefitinib resistance in lung cancer by activating ERBB3 signaling. *Science* 316: 1039-1043
- Ercan D, Xu C, Yanagita M, Monast CS, Pratilas CA, Montero J, Butaney M, Shimamura T, Sholl L, Ivanova EV *et al* (2012) Reactivation of ERK signaling causes resistance to EGFR kinase inhibitors. *Cancer Discov* 2: 934-947
- Fritz G, Brchetti C, Bahlmann F, Schmidt M, Kaina B (2002) Rho GTPases in human breast tumours: expression and mutation analyses and correlation with clinical parameters. *Br J Cancer* 87: 635-644
- Glorian V, Maillot G, Poles S, Iacovoni JS, Favre G, Vagner S (2011) HuR-dependent loading of miRNA RISC to the mRNA encoding the Ras-related small GTPase RhoB controls its translation during UV-induced apoptosis. *Cell Death Differ* 18: 1692-1701
- Herbst RS, Prager D, Hermann R, Fehrenbacher L, Johnson BE, Sandler A, Kris MG, Tran HT, Klein P, Li X *et al* (2005) TRIBUTE: a phase III trial of erlotinib hydrochloride (OSI-774) combined with carboplatin and paclitaxel chemotherapy in advanced non-small-cell lung cancer. *J Clin Oncol* 23: 5892-5899
- Lee HJ, Zhuang G, Cao Y, Du P, Kim HJ, Settleman J (2014) Drug resistance via feedback activation of Stat3 in oncogene-addicted cancer cells. *Cancer Cell* 26: 207-221
- Liu M, Tang Q, Qiu M, Lang N, Li M, Zheng Y, Bi F (2011) miR-21 targets the tumor suppressor RhoB and regulates proliferation, invasion and apoptosis in colorectal cancer cells. *FEBS Lett* 585: 2998-3005
- Mazieres J, Antonia T, Daste G, Muro-Cacho C, Berchery D, Tillement V, Pradines A, Sebti S, Favre G (2004) Loss of RhoB expression in human lung cancer progression. *Clinical cancer research : an official journal of the American Association for Cancer Research* 10: 2742-2750
- Mazieres J, Tovar D, He B, Nieto-Acosta J, Marty-Detraves C, Clanet C, Pradines A, Jablons D, Favre G (2007) Epigenetic regulation of RhoB loss of expression in lung cancer. *BMC Cancer* 7: 220
- Politi K, Zakowski MF, Fan PD, Schonfeld EA, Pao W, Varmus HE (2006) Lung adenocarcinomas induced in mice by mutant EGF receptors found in human lung cancers respond to a tyrosine kinase inhibitor or to down-regulation of the receptors. *Genes & development* 20: 1496-1510
- Sato N, Fukui T, Taniguchi T, Yokoyama T, Kondo M, Nagasaka T, Goto Y, Gao W, Ueda Y, Yokoi K *et al* (2007) RhoB is frequently downregulated in non-small-cell lung cancer and resides in the 2p24 homozygous deletion region of a lung cancer cell line. *International journal of cancer* 120: 543-551
- Shi C, Liang Y, Yang J, Xia Y, Chen H, Han H, Yang Y, Wu W, Gao R, Qin H (2013) MicroRNA-21 knockout improve the survival rate in DSS induced fatal colitis through protecting against inflammation and tissue injury. *PloS one* 8: e66814
- Sun G, Li H, Rossi JJ (2010) Sequence context outside the target region influences the effectiveness of miR-223 target sites in the RhoB 3'UTR. *Nucleic Acids Res* 38: 239-252

Tan Y, Yin H, Zhang H, Fang J, Zheng W, Li D, Li Y, Cao W, Sun C, Liang Y *et al* (2015) Sp1-driven up-regulation of miR-19a decreases RHOB and promotes pancreatic cancer. *Oncotarget* 6: 17391-17403

Wang S, Yan-Neale Y, Fischer D, Zeremski M, Cai R, Zhu J, Asselbergs F, Hampton G, Cohen D (2003) Histone deacetylase 1 represses the small GTPase RhoB expression in human nonsmall lung carcinoma cell line. *Oncogene* 22: 6204-6213

Yang Y, Ma Y, Shi C, Chen H, Zhang H, Chen N, Zhang P, Wang F, Yang J, Yang J *et al* (2013) Overexpression of miR-21 in patients with ulcerative colitis impairs intestinal epithelial barrier function through targeting the Rho GTPase RhoB. *Biochem Biophys Res Commun* 434: 746-752

Zeng Y, Zhang X, Kang K, Chen J, Wu Z, Huang J, Lu W, Chen Y, Zhang J, Wang Z *et al* (2016) MicroRNA-223 Attenuates Hypoxia-induced Vascular Remodeling by Targeting RhoB/MLC2 in Pulmonary Arterial Smooth Muscle Cells. *Sci Rep* 6: 24900

3rd Editorial Decision

09 November 2016

Thank you for the submission of your revised manuscript to EMBO Molecular Medicine. We have now received the enclosed reports from the referees that were asked to re-assess it. As you will see the reviewers are now globally supportive and I am pleased to inform you that we will be able to accept your manuscript pending the following final editorial amendments:

- 1) Please complete in full all sections of the provide author checklist. This will be published within the review process file alongside the paper.
- 2) Connected to the above, the manuscript must include a statement in the Materials and Methods identifying the institutional and/or licensing committee approving the experiments, including any relevant details (like how many animals were used, of which gender, at what age, which strains, if genetically modified, on which background, housing details, etc). We encourage authors to follow the ARRIVE guidelines for reporting studies involving animals. Please see the EQUATOR website for details: <http://www.equator-network.org/reporting-guidelines/improving-bioscience-research-reporting-the-arrive-guidelines-for-reporting-animal-research/>. Please make sure that all the above details are reported.
- 3) We encourage the publication of source data, particularly for electrophoretic gels and blots, with the aim of making primary data more accessible and transparent to the reader. Would you be willing to provide a PDF file per figure that contains the original, uncropped and unprocessed scans of all or at least the key gels used in the manuscript? The PDF files should be labeled with the appropriate figure/panel number, and should have molecular weight markers; further annotation may be useful but is not essential. The PDF files will be published online with the article as supplementary "Source Data" files. If you have any questions regarding this just contact me.
- 4) Every published paper includes a 'Synopsis' to further enhance discoverability. Synopses are displayed on the journal webpage and are freely accessible to all readers. They include a short standfirst as well as 2-5 one sentence bullet points that summarise the paper. Please provide the synopsis including the short list of bullet points that summarise the key NEW findings. The bullet points should be designed to be complementary to the abstract - i.e. not repeat the same text. We encourage inclusion of key acronyms and quantitative information. Please use the passive voice. Please attach this information in a separate file or send them by email, we will incorporate it accordingly. You are also welcome to suggest a striking image or visual abstract to illustrate your article. If you do please provide a jpeg file 550 px-wide x 400-px high.
- 5) We now mandate that ALL corresponding authors list an ORCID digital identifier. You may acquire one through our web platform upon submission and the procedure takes <90 seconds to complete. We also encourage co-authors to supply an ORCID identifier, which will be linked to their name for unambiguous name identification.
- 6) Please complete figures 2A and 4B with the origin boxes for the magnified insets.

7) I note that the quality of figure 6A is quite low and appears blurry. Please provide a better image as this could lead to problems when the production team tries to resize these images for the final manuscript.

8) Please combine all appendix figures, legends, supplementary materials in a single PDF

9) All figures should be presented in portrait format (rather than landscape).

Please submit your revised manuscript within two weeks. I look forward to seeing a revised form of your manuscript as soon as possible.

***** Reviewer's comments *****

Referee #1 (Comments on Novelty/Model System):

The authors improved the quality of their study by analysing more patients and extending the study to additional variables. Overall, the revised manuscript and new data appropriately answer my critique.

Referee #2 (Remarks):

The authors have now addressed my comments satisfactorily.

4th Revision – Author's Response

21 November 2016

1) Please complete in full all sections of the provide author checklist. This will be published within the review process file alongside the paper.

Complete author checklist is now provided.

2) Connected to the above, the manuscript must include a statement in the Materials and Methods identifying the institutional and/or licensing committee approving the experiments, including any relevant details (like how many animals were used, of which gender, at what age, which strains, if genetically modified, on which background, housing details, etc). We encourage authors to follow the ARRIVE guidelines for reporting studies involving animals. Please see the EQUATOR website for details: <http://www.equator-network.org/reporting-guidelines/improving-bioscience-research-reporting-the-arrive-guidelines-for-reporting-animal-research/>. Please make sure that all the above details are reported.

Statements that identify the institutional committees approving this study are already mentioned in the Materials and Methods section. For animal study, approval was received from the Claudius Regaud Institute Animal Ethics Committee (# ICR-2009-021). For the use of human tissue samples, study was approved by the Ethics of Human Research Committee at the Pathology Department, Toulouse Hospital, France.

Information about the mice used in this study (including strain, age, housing conditions, genetic modifications and references about the origins of the strains) is detailed in the corresponding Materials and Methods section; the number of animals used for each experiment is detailed in the legend of corresponding figure.

3) We encourage the publication of source data, particularly for electrophoretic gels and blots, with the aim of making primary data more accessible and transparent to the reader. Would you be willing to provide a PDF file per figure that contains the original, uncropped and unprocessed scans of all or at least the key gels used in the manuscript? The PDF files should be labeled with the appropriate figure/panel number, and should have molecular weight markers; further annotation may be useful but is not essential. The PDF files will be published online with the article as supplementary "Source Data" files. If you have any questions regarding this just contact me.

Source data for all the Western Blots presented in the study are provided as PDF files. Each PDF file corresponds to one figure excepted for Appendix figures for which original blots were pooled in a single PDF file.

4) Every published paper includes a 'Synopsis' to further enhance discoverability. Synopses are displayed on the journal webpage and are freely accessible to all readers. They include a short standfirst as well as 2-5 one sentence bullet points that summarise the paper. Please provide the synopsis including the short list of bullet points that summarise the key NEW findings. The bullet points should be designed to be complementary to the abstract - i.e. not repeat the same text. We encourage inclusion of key acronyms and quantitative information. Please use the passive voice. Please attach this information in a separate file or send them by email, we will incorporate it accordingly. You are also welcome to suggest a striking image or visual abstract to illustrate your article. If you do please provide a jpeg file 550 px-wide x 400-px high.

Standfirst: High RHOB levels in EGFR-mutated lung tumors predict resistance to EGFR-TKI therapy, and combining AKT inhibition could be a therapeutic strategy to restore drug sensitivity in RHOB positive patients.

Bullet 1: High RHOB expression levels are associated with resistance to EGFR-TKI in lung cancer cell lines and patients harboring EGFR activating mutation and in EGFR^{L858R}-driven lung cancer mouse model.

Bullet 2: Median PFS after EGFR-TKI treatment is 15.3 months for patients with low RHOB tumor levels and 5.6 months for patients with high RHOB levels.

Bullet 3: RHOB induces EGFR-TKI resistance by preventing AKT inhibition.

Bullet 4: AKT inhibition with the new specific inhibitor Ipatasertib (G594) reverses RHOB-induced resistance to erlotinib *in vitro* and *in vivo*.

5) We now mandate that ALL corresponding authors list an ORCID digital identifier. You may acquire one through our web platform upon submission and the procedure takes <90 seconds to complete. We also encourage co-authors to supply an ORCID identifier, which will be linked to their name for unambiguous name identification.

Pr. Julien Mazieres' ORCID digital identifier: 0000-0002-5921-7613

6) Please complete figures 2A and 4B with the origin boxes for the magnified insets.

Origin boxes for the magnified insets have been inserted.

7) I note that the quality of figure 6A is quite low and appears blurry. Please provide a better image as this could lead to problems when the production team tries to resize these images for the final manuscript.

A better quality image is now provided.

8) Please combine all appendix figures, legends, supplementary materials in a single PDF

This has been done.

9) All figures should be presented in portrait format (rather than landscape).

This has been changed.

Corresponding Author Name: Gilles Favre

Manuscript Number: EMM-2016-06646-V2